# Refinement of the ice absorption spectrum in the visible using radiance profile measurements in Antarctic snow

Ghislain Picard[1,2], Quentin Libois[1,a], and Laurent Arnaud[1]

[1]UGA / CNRS, Laboratoire de Glaciologie et Géophysique de l'Environnement (LGGE) UMR 5183, Grenoble, F-38041, France
[2]ACE CRC, University of Tasmania, Private Bag 80, Hobart, TAS 7001, Australia
[a]now at: Department of Earth and Atmospheric Sciences, Université du Québec à Montréal (UQAM), Montréal, Canada

*Correspondence to:* Ghislain Picard (ghislain.picard@univ-grenoble-alpes.fr)

**Abstract.** Ice is a highly transparent material in the visible. According to the most widely used database (IA2008, Warren and Brandt, 2008), the ice absorption coefficient reaches values lower than $10^{-3}\,\mathrm{m^{-1}}$ around 400 nm. These values were obtained from a spectral radiance profile measured in a single snow layer at Dome C in Antarctica. We reproduced this experiment using an optical fiber inserted in the snow to record 56 profiles from which 70 homogeneous layers were identified. Applying the

same estimation method on every layer yields 70 ice absorption spectra. They present a significant variability but absorption coefficients are overall larger than IA2008 by one order of magnitude at 400-450 nm. We devised another estimation method based on Bayesian inference that treats all the profiles simultaneously. It reduces the statistical variability and confirms the higher absorption, around $2 \times 10^{-2}\,\mathrm{m^{-1}}$ near the minimum at 440 nm. We explore potential instrumental artifacts by developing a 3D radiative transfer model able to explicitly account for the presence of the fiber in the snow. The simulation shows

that the radiance profile is indeed perturbed by the fiber intrusion, but the error on the ice absorption estimate is not larger than a factor of 2. Hence this is insufficient to explain the difference between our new estimate and IA2008. The same conclusion applies regarding the plausible contamination by black carbon or dust, concentrations reported in the literature are insufficient. Considering the large number of profiles acquired for this study and other estimates from the Antarctic Muon and Neutrino Detector Array (AMANDA), we nevertheless estimate that ice absorption values around $10^{-2}\,\mathrm{m^{-1}}$ at the minimum are more

likely than under $10^{-3}\,\mathrm{m^{-1}}$. A new estimate in the range 400–600 nm is provided for future modeling of snow, cloud, and sea-ice optical properties. Most importantly, we recommend that modeling studies take into account the large uncertainty of the ice absorption coefficient in the visible and that future estimations of the ice absorption coefficient should also thoroughly account for the impact of the measurement method.

## 1 Introduction

The spectral absorption coefficient of ice is one of the primary variables controlling the reflectance and transmittance of snow covered surfaces, sea ice and ice clouds. It is responsible for the main spectral features of snow albedo: in the visible, the albedo is close to 1 and it smoothly decreases in the near-infrared with significant absorption bands around 1030, 1300, and 1550 nm (Warren and Wiscombe, 1980). These features are critical for the surface radiative budget in general, and snow-albedo

feedbacks in particular (Qu and Hall, 2007; Box et al., 2012; Picard et al., 2012). The subsurface absorption of solar radiation is also of uttermost importance for the surface energy budget (Brandt and Warren, 1993; Kuipers Munneke et al., 2009; Flanner and Zender, 2005) and for photochemistry (Domine et al., 2008). In marine environments, the light transmitted through sea-ice controls phytoplankton and algae growth (Perovich, 1993; Melbourne-Thomas et al., 2015) and contributes to ice melt and
warming of the ocean superficial layer (Ehn et al., 2008; Katlein et al., 2015).

Knowledge of the ice absorption coefficient is essential but some significant uncertainties remain in the lower range of the microwaves (e.g. Mätzler, 1998; Leduc-Leballeur et al., 2015; Macelloni et al., 2016), UV and visible. In the two latter domains, numerous updates have covered 3 orders of magnitude in the last decades, illustrating these uncertainties (e.g. Grenfell et al., 1981; Warren, 1984; Askebjer et al., 1997; Ackermann et al., 2006; Warren and Brandt, 2008). In general the trend has been
towards lower absorption values, and the most recent and widely used compilation by Warren and Brandt (2008) suggests values as low as $6 \times 10^{-4}\,\mathrm{m}^{-1}$ at the frontier between the UV and the visible (this ice absorption dataset is called IA2008 hereinafter). This corresponds to light intensity decreasing by approximately a factor of 2 per kilometer in pure ice. A related debate concerns the position of minimum absorption, which in general has shifted towards shorter wavelengths with successive updates.
Estimation of the absorption coefficient from light transmission measurements directly in ice slabs proved to be difficult for several reasons. First, to measure significant decrease due to the absorption, the path length in the slab needs to be large compared to the inverse of the absorption coefficient, that is of the order of kilometers. Second, the residual scattering and reflections in the slab reduce the transmission and can be misinterpreted as absorption (Askebjer et al., 1997). These processes need to be either minimized and neglected, or accurately estimated and corrected. Last, it is hard to prepare or find large
quantities of perfectly pristine ice on Earth, because very tiny amounts of light absorbing impurities (LAI) embedded in ice are sufficient to dominate the total absorption coefficient of the mixture. All these problems are most prominent at wavelengths where the absorption is low, explaining why the uncertainties remain large in this spectral range. While controlling the purity of ice is a general issue that seems difficult to overcome, an alternative has been proposed by Warren et al. (2006) to address the scattering and path length issue. Instead of limiting scattering, this original approach measures light extinction in snow, where
scattering is dominant. The benefit is a large path length compared to the size of the experiment. For instance, most photons found at $20\,\mathrm{cm}$ depth in snow have traveled several meters since they entered the snowpack at the surface. A similar approach is used in spectrometers or advanced interferometers where light is reflected forth and back on mirrors (e.g. Romanini et al., 2006; Abbott et al., 2016). However, contrary to using mirrors with high and known reflectance, snow scattering is in general high but depends on unknown snow micro-structure properties. Hence the path length is large but difficult to accurately estimate. The
method proposed by Warren et al. (2006) overcomes this issue by assuming the ice absorption at $\lambda_1 = 600\,\mathrm{nm}$ is known and demonstrates that the absorption coefficient at any other wavelength $\lambda_2$ can be simply inferred from the ratio of the asymptotic flux extinction coefficients (AFEC) at both wavelengths and the known absorption at $\lambda_1$. The AFEC – that is the inverse of the $e$-folding depth (e.g. France et al., 2011) – is obtained from the gradient of the logarithm of the radiance profile (log-radiance hereinafter). Warren et al. (2006) measured radiance profiles by recording spectral radiance at several depths (up to $1.35\,\mathrm{m}$) in
the snow using an optical fiber progressively and vertically inserted. The measurements were taken in a snowpit near Dome C

in Antarctica where two homogeneous layers were identified based on the stratigraphy and exploited to retrieve the absorption coefficient in the range 350–600 nm. IA2008 corresponds to the deepest of these two layers (called Layer C) which showed a weaker extinction. This yielded a significantly lower estimate of ice absorption compared to the previous compilation (Warren, 1984) and other contemporaneous estimates (e.g. Ackermann et al., 2006). Warren et al. (2006) noted "This result is tentative; we are skeptical of our low $k_{ice}$ because (. . . )" and "It would be good to repeat this experiment at a site much farther from the station than our 2 km site, where even the surface snow would be as clean as layer C". These statements motivated the present study.

In this paper, we aim at refining the ice absorption coefficient using a method similar to Warren et al. (2006) and by addressing three critical issues. First, the number of investigated snowpits was limited in Warren et al. (2006). To reproduce the experiment and permit a statistical analysis, we collected 56 profiles. Moreover, we devised a Bayesian estimation method to retrieve the absorption from this large dataset. This provides not only the most likely ice absorption spectrum but also gives the uncertainty related to the estimation process and observation errors. Second, it is possible that the insertion of the optical fiber in the snow disturbs the radiance field near the fiber tips where the light ultimately enters the spectrometer (Light et al., 2008; Petrich et al., 2012). Since the fiber housing is likely to have a lower reflectance than snow, the housing absorbs more than the surrounding snow. Intuitively, this could increase the apparent AFEC and in turn result in an over-estimation of the absorption. Following Light et al. (2008) and Petrich et al. (2012) who investigated similar disturbances in the case of sea-ice transmission measurements, we performed 3D radiative transfer simulations where the fiber is explicitly modeled. The results underscore the conditions where the gradient estimation is only moderately affected by the instrument. Third, the impurities content in snow has to be negligible but the contamination by local activities around polar stations has been shown to reach significant levels (Grenfell et al., 1994). Measurements used in Warren et al. (2006) were collected in 2005 about 2 km from Dome C station and only the layer C under 40 cm depth was found to be clean enough. Since then, the Concordia station has been operating all year round which has likely increased the contamination in extent and in depth in the snow. To minimize this issue, our measurements were collected at distinct locations and at great distances (3–25 km) from the station including downwind and upwind sites.

The paper is organized as follows: Section 2 describes the radiance profile measurements, the original and the Bayesian methods to estimate the ice absorption, and the 3D radiative transfer model used to simulate the presence of the fiber within the snow. Section 3 presents the ice absorption estimation results and explores the experimental artifacts using the model simulations. The discussion in Section 4 compares our new estimate to previous ones.

## 2 Materials and method

### 2.1 Measurements of radiance profile

Solar Extinction in Snow (SOLEXS) is a device to measure the rate of radiance decrease in snow (Libois et al., 2014b). It consists of an optical fiber fitted in a white-painted (color RAL 9003) stainless steel rod (total diameter 10 mm) which is vertically inserted in snow into a hole with same diameter dug beforehand (Fig. 1). The vertical position is measured using

a magnetic coding rule with 1 mm resolution. The fiber is connected to an Ocean Optics MayaPro spectrometer covering the spectral range 300–1100 nm with 3 nm resolution. While the fiber is manually displaced continuously in the hole, the spectrum is recorded every 5 mm, during descent and ascent. The integration time of the spectrometer is automatically adjusted to optimize the signal and ranges from the minimum possible (7 ms) to a maximum of 1000 ms. Although longer integration

times are possible to take measurements at greater depths where the light intensity is lesser, it would require to move the fiber increasingly slower ($< 5\,\mathrm{mm\,s^{-1}}$), otherwise the vertical resolution of the profile would be degraded. Additional factors limit the depth of measurements, such as the increasing influence of shadows on the surface. SOLEXS structure is made of thin elements (Fig. 1) and is aligned with respect to the sun to minimize shadows. The operator is about 1 m from the fiber insertion point and has the sun on his/her side so the shadow lays away from the insertion point. As a rule of thumb, the influence on the

radiation field in depth starts at half the distance of the disturbance at the surface, so that in our case the profiles up to 50 cm are considered unaffected (Petrich et al., 2012). Given the range of integration time and the dynamic range of the spectrometer (1:500 according to the manufacturer), SOLEXS is able to cover more than 5 decades of radiance change near the maximum of irradiance (i.e. 500 nm). Measuring a single two-way 50-cm deep profile takes less than 2 min once the setting is deployed. To monitor the variations of incident radiation during this time, we place a photosensor on the surface which records broadband

incident irradiance for each spectrum acquisition. Profiles acquired during large variations of illumination (typically over 3%) are immediately discarded in the field. The reproducibility between the ascending and descending measurements provides an indication of the overall quality and illumination stability.

Similar radiance profilers were used by Warren et al. (2006) and Light et al. (2008). The main difference is the light collector. SOLEXS measures light coming from the sides using translucent low-absorption materials (teflon, Figure 1), while Warren

et al. (2006) used a bare fiber (protected by a sapphire window) looking downward and Light et al. (2008) used an upward-looking diffuser. We have performed experiments with the bare fiber configuration and found similar rates of decrease to those with the side looking configuration. This is expected because the radiance field is essentially isotropic at depth greater than a few centimeters. It is worth noting that these systems do not measure absolute irradiance because the angular response of the light collector is not precisely known and is not strictly cosine (or isotropic). This limitation is not critical for the present study

which only uses the derivative of the profiles in log scale and relies on high scattering in snow so that all radiative quantities (irradiance, actinic flux, intensity in any direction, ...) decrease at the same rate. Nevertheless, for the sake of simplicity, we use the term "profile of log-radiance" throughout this article.

SOLEXS is accompanied by a dedicated software with graphical user interface to control, visualize and annotate the acquisitions directly in the field, and a post-processing library to produce the profiles. The following processing steps are applied:

1) subtraction of the dark current and normalization by the integration time as in (Picard et al., 2016), 2) precise estimation of the depth at which each spectrum has been acquired using the small difference of timestamps between the depth and spectrum acquisitions, 3) normalization by the photosensor current.

## 2.2 Measurements around Dome C

A total of 56 profiles have been acquired during the summer seasons 2012-2013 and 2013-2014 between 3 and 25 km from Concordia station and in different directions. The prevailing winds come from the South-South-West sector around Dome C (Champollion et al., 2013). On January 7, 2014, we specifically organized a transect on the route toward the South GLACIO-CLIM site (Genthon et al., 2015) in the upwind direction. 33 profiles were collected the same day at 4, 6, 8 and 10 km from the station. All the profiles at 400, 500, 600 and 700 nm are shown in Supplementary Figure 1. Following Warren et al. (2006), we visually determined homogeneous zones in these profiles based on the linearity of the log-radiance. To limit the impact of this subjective selection, the three authors of the study performed independent assessments, and only where at least two authors agreed, the zone was retained. Data within 8 cm from the surface were systematically discarded because of the perturbation by the rod (see Section 3.3). The dataset includes a total of 70 homogeneous zones (gray shade in Figure S1).

## 2.3 Estimation of the ice absorption

According to radiative transfer theory, the decrease of intensity (or diffuse radiance, actinic flux, etc) $I$ with depth $z$ in an homogeneous zone (far from any interface) of a turbid or diffusive horizontally layered medium follows an exponential form:

$$I(z, \lambda) = I(z = 0, \lambda) e^{-k_e(\lambda)z}, \tag{1}$$

where $\lambda$ is the wavelength and $k_e(\lambda)$ is the AFEC. The latter can be easily estimated from observed profiles by using linear regression of $\log I(z, \lambda)$. Theoretically $k_e(\lambda)$ is related to intrinsic optical properties of the snow (a.k.a single scattering properties). For instance, under the $\delta$-Eddington approximation (Joseph et al., 1976; Wiscombe and Warren, 1980) which applies well to snow in the visible where scattering is much stronger than absorption, it reads:

$$k_e(\lambda) = \sigma_e \sqrt{3(1 - g\omega(\lambda))(1 - \omega(\lambda))}, \tag{2}$$

where $\omega$, $\sigma_e$ and $g$ are the single scattering albedo, extinction coefficient and asymmetry factor of the snow, respectively. It has been shown in the case of pure snow (i.e. made solely of pure ice) that the factor $(1 - \omega(\lambda))$ under the square root is proportional to the ice absorption coefficient $\gamma(\lambda)$ (Warren et al., 2006) while the other factors on the right hand side ($\sigma_e$ and $(1 - g\omega(\lambda))$) mostly depend on the micro-structure, with very little dependence on the wavelength (Wiscombe and Warren, 1980). This can be analytically demonstrated following Kokhanovsky and Zege (2004) which relate the optical single scattering properties to the specific surface area (SSA) and density $\rho$ as follows:

$$\sigma_e = \sigma_s + \sigma_a = \frac{\rho \text{SSA}}{2} \qquad 1 - \omega = \sigma_a/\sigma_e = \frac{2B\gamma_{\text{ice}}}{\rho_{\text{ice}}\text{SSA}} \qquad \gamma_{\text{ice}} = \frac{4\pi}{\lambda} n_i \tag{3}$$

where $\rho_{\text{ice}} = 917 \text{ kg m}^{-3}$ is the ice density and $B = 1.6$ the absorption enhancement parameter which has very little dependence to the wavelength (Kokhanovsky and Zege, 2004; Libois et al., 2014b). $\sigma_s$ and $\sigma_a$ are the scattering and absorption coefficients of the snow, and $n_i$ the imaginary part of ice refractive index. These equations assume that snow grains are randomly oriented particles, weakly absorbing and the real part of refractive index is nearly independent on the wavelength (Libois et al., 2013).

Combining Equations 2 and 3, the AFEC can be explicitly written as a function of snow geometric properties:

$$k_e(\lambda) = \sqrt{\gamma_{\text{ice}}(\lambda)}\sqrt{\frac{3B\text{SSA}\rho^2(1-g)}{4\rho_{\text{ice}}}}, \tag{4}$$

where the approximation $(1-g\omega) \approx (1-g)$ has been used. The interesting feature in these two equations for our application is the proportionality of $k_e(\lambda)$ to $\sqrt{\gamma_{\text{ice}}(\lambda)}$. The scaling coefficient (rightmost square root term) is independent of the wavelength

and only depends on snow geometric properties.

To derive the ice absorption, the first method used here follows Warren et al. (2006) (WBG method). It assumes that the ice absorption is known at $\lambda_0 = 600\,\text{nm}$ and estimates $k_e(\lambda)$ and $k_e(\lambda_0)$ from the log-radiance profiles to deduce the ice absorption at any other wavelength $\lambda$ using:

$$\gamma_{\text{ice}}(\lambda) = \gamma_{\text{ice}}(\lambda_0)\left(\frac{k_e(\lambda)}{k_e(\lambda_0)}\right)^2. \tag{5}$$

The second method follows the same physical principles but accounts in addition for the uncertainties in the observations and the variable thickness of the zones. It uses Bayesian inference (BAY) to deduce not only the most likely $\gamma_{\text{ice}}(\lambda)$ but also its uncertainty. The proportionality and the estimation of $k_{e\,(n,i)}$ are translated into a statistical generalized linear model as follows:

$$k_{e\,(n,i)} = \alpha_n\sqrt{\gamma(\lambda_i)}, \tag{6}$$

$$\log I_{\text{est}}(\mathbf{z}_n, \lambda_i) = -k_{e\,(n,i)}\mathbf{z}_n + b_{n,i}, \tag{7}$$

$$\log I_{\text{obs}}(\mathbf{z}_n, \lambda_i) \sim \mathcal{N}(\log I_{\text{est}}(z_n, \lambda_i), \sigma^2), \tag{8}$$

where the index $i$ runs on the wavelengths and $n$ on the homogeneous zones. $\log I_{\text{obs}}(\mathbf{z}_n, \lambda_i)$ is the log-radiance observed in zone $n$ at all the depths $\mathbf{z}_n$. $\sigma$ measures the observation errors which, for sake of simplicity, are assumed identical for all the measurements. All the variables (excepted $\mathbf{z}_n$) are random variables and follow a prior distribution that should reflect the

degree of "expert knowledge", that is the knowledge we have before using any observations. Here, we consider that $\alpha_n$, $\sigma$ and $b_{n,i}$ are unknown (i.e. highly uncertain) so we choose uninformative prior (i.e; wide distribution):

$$\sigma \sim |\mathcal{N}(0, 1.0)|, \tag{9}$$

$$\alpha_n \sim \mathcal{U}(5, 40), \tag{10}$$

$$b_{n,i} \sim \mathcal{N}(\log(I_0), 6), \tag{11}$$

where $I_0$ is the typical intensity measured at the surface. $\mathcal{U}$ and $\mathcal{N}$ are the uniform and normal distributions respectively. Prior $\alpha_n$ distribution spans a very large range from $5\,\text{m}^{1/2}$ to $40\,\text{m}^{1/2}$. This largely encompasses typical Dome C snow. To give an idea, snow with SSA as low as $8\,\text{m}^2\,\text{kg}^{-1}$ and any density larger than $130\,\text{kg}\,\text{m}^{-3}$ fits in this range. For high SSA of $50\,\text{m}^2\text{kg}^{-1}$, any density between 60 to $400\,\text{kg}\,\text{m}^{-3}$ is suitable. We checked that the posterior distributions of the variables are much narrower which means that the results are insensitive to these choices. In contrast, the prior distribution of $\gamma(\lambda_i)$

needs to be informative to some degree because the method can not predict absolute absorption coefficient (see Equation 5). The method WBG is indeed equivalent to considering that the prior of absorption is certain at $\lambda_0 = 600\,\text{nm}$ and ignorance at any other wavelengths. Here, we consider that IA1984 (Warren, 1984) and IA2008 are two extreme estimates. Hence the most likely prior value is taken as the average of IA1984 and IA2008 and the uncertainty related to the difference between IA1984 and IA2008 through a linear function. This function has an offset to represent the uncertainty when the difference is null, that is for $\lambda > 600\,\text{nm}$. Note that since $\gamma(\lambda)$ spans several orders of magnitude in the visible, we compute the difference and average in logarithm scale and choose a log-normal distribution for the prior. Formally it reads:

$$\log \gamma_{\text{ice}}(\lambda_i) \sim \mathcal{N}\left(\frac{\log \gamma_{\text{ice}}^{(2008)}(\lambda_i) + \log \gamma_{\text{ice}}^{(1984)}(\lambda_i)}{2}, |\log \gamma_{\text{ice}}^{(2008)}(\lambda_i) - \log \gamma_{\text{ice}}^{(1984)}| + \sigma_0\right) \tag{12}$$

where $\sigma_0$ is the standard deviation (in logarithm scale) when the difference between both datasets vanishes. Figure 2 shows this prior distribution along with IA1984 and IA2008 data. The spread is very large at wavelengths shorter than $600\,\text{nm}$ (weakly informative prior), with possible absorption coefficient extending beyond the range of the 1984 and 2008 datasets. On the other hand, the uncertainties at the longer wavelengths is determined by $\sigma_0 = 0.23$ which is equivalent to $\pm 25\%$ relative uncertainty.

The statistical problem so stated is solved for the 70 homogeneous zones and 29 wavelengths from $320\,\text{nm}$ to $880\,\text{nm}$ ($20\,\text{nm}$ step). Some deep zones have no data for the longer wavelengths because the measurements are under the noise level. The problem is huge with a total of 1417 unknowns and 40261 observations. The computation of the posterior distributions is performed using a Markov-Chain Monte Carlo sampler called No-U-Turn Sampler (NUTS; Hoffman and Gelman, 2014). We use the software package pymc3 (https://github.com/pymc-devs/pymc3) which is easy to use and efficient even without expert tuning. We run 8 independent chains from which 20,000 samples are drawn. The first half is discarded (burn in) to allow the sampler to converge and one sample every 100 is taken from the second half (thinning) to avoid auto-correlation between successive samples in the chains. The resulting 800 samples are used to produce statistics presented in the next section. For the sake of readability, plots use only the first 100 samples.

## 2.4 Snow 3D radiative transfer

Both estimation methods (WBG and BAY), whatever their degree of sophistication, consider that the errors are independent random noise with zero mean, that is, that the observations are non-biased measures of the profile of radiance (at least proportional to the radiance). To test this assumption, we investigate the impact of the presence of the rod in the snow. To this end, the 3D radiative transfer model "Monte Carlo modeling of light transport in multi-layered tissues" (MCML, Wang et al., 1995) has been adapted to model snow scattering and absorption, SOLEXS rod and optionally air space around the rod as depicted in Figure 3. The original model takes as input the scattering $\sigma_s$ and absorption $\sigma_a$ coefficients and asymmetry factor $g$ in every layer that extends horizontally infinitely. It solves the radiative transfer equation by launching $N$ light rays from the source and following their trajectory and intensity. The trajectory is a random walk with step length governed by a Poisson random variable parametrized by the extinction coefficient $\sigma_e = \sigma_s + \sigma_a$ (Wang et al., 1995). Change of direction is governed by the Henyey-Greenstein phase function with asymmetry factor $g$. At each step, the intensity is decreased by the factor $\sigma_a/\sigma_e$ (a.k.a. single scattering co-albedo). Rays with an intensity less than a specified threshold are to be discarded. However, since abruptly

discarding all the rays would result in a small bias (smaller than the threshold), a process known as Russian *Roulette* (section 3.9 of Wang et al., 1995) is applied. A small proportion of the rays (typically $p = 10\%$) is randomly chosen, their intensity is multiplied by $1/p$ and they are re-injected in the normal process of propagation. All the other rays are discarded. The threshold was set to $10^{-5}$ (the initial intensity of rays is 1) which ensures bias much smaller than SOLEXS dynamic range.

The rod housing the optical fiber is modeled by a cylinder (radius $r_{rod}$ and depth $z_{rod}$) with a given albedo ($\omega_{rod}$). The rays hitting the cylinder are reflected back in a random direction (Lambertian reflection) and their intensity is decreased by the factor $1 - \omega_{rod}$. The hole (air void) around the rod is also modeled as a cylinder (radius $r_{hole}$ and depth $z_{hole}$) in which rays propagate in straight line and are not attenuated.

  The model records the total absorption as a function of depth and radius (the calculation is done in three-dimension, but the
recording assumes cylindrical symmetry). In addition, it records rays escaping the medium by the upper interface (i.e. reflected rays) and lower interface (i.e. transmitted rays). We extend the original code, which only considers a narrow beam source, to support two modes of operations. First mode aims to simulate infinitely large source (like a diffuse sky). For this, rays are launched from a circular area of radius $r_{source}$ located above the surface which is wide compared to the typical propagation length of the rays (1 m in our case). A too large radius is suboptimal because rays launched and propagating too far from the
area of interest (i.e. the rod) are useless for the calculation. In other words, any ray that does not hit the rod at least once is uninteresting because it contributes to the solution of the 1-D plane parallel medium radiative transfer equation which is well know and can be solved by more efficient methods compared to Monte-Carlo. For this reason we developed a second mode aimed at optimizing the calculations when only the rays that enter the collector are of interest. As launching rays from a wide source and excepting they will enter the (small) collector is highly improbable, we use the inverse principle in optics
(equivalent to inverting time in the radiative transfer equation), launch rays from the collector and accumulate the intensity of those reaching the surface. Hence all rays contribute to the calculation, which allows convergence within 1% error with $N = 10,000$ rays only. This paper only uses the second mode.

  The optical single scattering properties of snow ($\sigma_s$ and $\sigma_a$) are calculated from the SSA and density using Equation 3. The asymmetry factor used for the phase function is 0.86 (Libois et al., 2014b). $\omega_{rod}$ is set to 0.9 for all the wavelengths for default
simulations based on our reflectance measurements of the paint.

## 3 Results

This section presents the estimation of ice absorption on our dataset using WBG and BAY methods. The 3D radiative transfer model is then used to study the influence of the insertion of the rod in the snow.

### 3.1 Estimation of the ice absorption using least-square linear regression

The WBG method is first applied to one of the homogeneous zones for illustration. We selected the profile 25kmE_1 between 20 and 30 cm depth which is described and used in Libois et al. (2014b). The results in terms of ice absorption are shown in Figure 4 as a function of the wavelength for the range $360 - 660$ nm. The spectrometer is not sensitive enough outside this range

(for 20–30 cm depths). In contrast, within this range, the statistical error (95% confidence interval in gray shade) is very small, which indicates the profiles have little noise and the estimation method is robust. The results show a good agreement between our WBG estimate, IA1984 and IA2008 at wavelengths larger than 550 nm. Even though this could be partially explained by the methodology – which uses a reference wavelength at 600 nm – this result confirms that the ice absorption is relatively well known at the longest wavelengths of the visible domain. In contrast, at shorter wavelengths we found a large discrepancy exceeding one order of magnitude between our estimate and both IA1984 and IA2008. The wavelength of minimum absorption also significantly varies, 390, 430 and 470 nm respectively for IA2008, our estimate and IA1984. No conclusion in favor of any dataset can be drawn at this stage.

We applied the WBG method individually to all the homogeneous zones which yields 70 ice absorption spectra (Figure 5). Most spectra follow the same general trend as the previous example, featuring a slight decrease from UV to a minimum reached in the visible around 400–450 nm and an increase at longer wavelengths. The scatter is relatively small at long wavelengths (550 nm and longer) but reaches nearly 1 order of magnitude near the absorption minimum. A few spectra seem to be outsiders and/or noisy. This is probably because the zone thickness and measurement quality is variable throughout the dataset. The quality depends on several factors, including the depth of the zone, snow homogeneity, quality of the hole, etc. It is therefore incorrect to interpret the set of curve as representative of the uncertainty. The different curves are not equiprobable. Nevertheless, none of the spectra reaches either IA1984 or IA2008 in the range 320 – 500 nm. We also note that most values at wavelengths longer than the reference (600 nm) are slightly lower than IA1984 and IA2008. A first conclusion of this work is that using the same method as Warren et al. (2006) on a larger set of observations, we obtain very variable ice absorption and no agreement with the other datasets at wavelengths shorter than about 550 nm.

While the analysis and selection of this set of spectra could be refined, this is not the route we have chosen. The Bayesian inference method is more powerful to deal with the heterogeneity of the data quality, requires less assumption on the error distributions, and is inherently able to provide the best estimate weighted by the quality of each individual members of this set.

## 3.2 Estimation of the ice absorption using Bayesian inference

The BAY method is applied simultaneously to all the homogeneous zones and wavelengths. It yields a set of 800 absorption spectra drawn from the posterior. A subset of 100 spectra is shown in Figure 6. In this case, all the curves are equiprobable so that the gray density represents the posterior of the ice absorption value at each wavelength, i.e. an indication of the uncertainty due to observation errors and estimation process. The spread of the gray area appears to be very limited compared to the prior in Figure 2, which confirms that the prior was not or little informative and conversely that the observations strongly constrain the ice absorption estimates. Moreover, the uncertainty range is negligible in comparison with the discrepancy to IA1984 or IA2008 at wavelength shorter than about 500 nm. The curve set is comparable to the estimate obtained with WBG on the profile 25kmE_1 and to the average of all the zones in Figure 5, showing that the estimation method, either BAY and WBG, has not a major influence on the conclusion. All the results with our data converge to a minimum absorption around $2 \times 10^{-2} \, \mathrm{m}^{-1}$ located in the range 400–450 nm.

The influence of site where the profile have been measured is investigated by applying the BAY method on groups of profiles based on the distance and direction to the station (Figs. 7 and 8). It is worth noting that the number of homogeneous zones (and observations) in each subset is very different which calls a cautious interpretation. Overall the inter-group differences are small compared with the discrepancy with IA1984 or IA2008. However, they are in some cases significant compared to the posterior distribution width, which means these differences are statistically significant. The most distant and southward, upwind, sites exhibit lower absorption than those close to the station or in the north, downwind direction.

## 3.3 Snow MCML simulations

### 3.3.1 Ideal homogeneous snowpack

Snow MCML is used to evaluate the impact of the rod on SOLEXS measurements. We first consider a homogeneous semi-infinite snowpack with typical values for Dome C snow (SSA= $30\,\mathrm{m^2 kg^{-1}}$ and $\rho = 350\,\mathrm{kg\,m^{-3}}$). Figure 9a shows the profiles of relative radiance calculated by i) a standard 1-D plane parallel two-stream radiative transfer model (TARTES, Libois et al., 2014b), ii) snow MCML without the rod (called ideal case hereinafter) and iii) with a $10\,\mathrm{mm}$-diameter rod having 90% albedo (without air gap). Relative radiance is calculated by normalization at $z$=-1 cm for each simulation. This depth was chosen instead of the surface $z$=0 because TARTES outputs include the direct (unscattered) incident radiation which vanishes rapidly with depth whereas our MCML inverse mode calculation does not. All these simulations use IA2008 for the ice absorption. The difference obtained between TARTES and MCML is very small which confirms the equivalence between the two different models.

In the presence of the rod (square symbols in Fig. 9a), the radiance decreases much more sharply than in the ideal case in the horizon extending from the surface to about 7–10 cm depth. Under this horizon, the profile becomes nearly parallel to that of the ideal case which suggests the log-radiance gradients are almost equal. To better visualize the difference between the profiles we computed their ratio and plotted it in linear scale (Fig. 9b. Note that the simulations at $700\,\mathrm{nm}$ are noisy because the signal is weak). This ratio equals 1 for the ideal case (no rod effect) and it decreases as the radiance is affected by the rod presence. In addition to the decrease close the surface that is also clearly visible in log scale, we observe a slight decrease under the horizon 7–10 cm, from around 0.76 to 0.66 at 50 cm, which highlights that the gradients are close but not equal. Another interesting result is that the effect of the rod is relatively independent of the wavelength.

The influence of the rod can be explained as follows: with an albedo of 0.90, the rod absorbs more than snow whose single scattering albedo is larger than 0.99 at any wavelength in the range considered here. Hence, as the rod is inserted in the snow, the probability that a ray hits the rod before being captured by the optical fiber increases. Inserting the rod can be compared in a first approximation to adding light-absorbing impurities. The radiance decreases more sharply than it does in the ideal case because of the increasing additional absorption. However, while impurities are usually dispersed in horizontal layers in the snow, the rod is a highly localized and concentrated absorber. This difference has strong implications because rays propagating from the surface to the tip of the fiber do not necessarily interact with the rod. In other words, there are many ray trajectories from the surface to the fiber tip that do not touch the rod. Furthermore, considering only the rays that reach the fiber tip – and

that are ultimately measured by SOLEXS – it is evident that hitting the lower part of the rod is more likely than the upper one. This is due to dilution in the 3D space yielding a probability of interaction with the rod decreasing as the square power of the distance from the fiber tip. It implies in practice that only the "lower" part of the rod significantly contributes to the rod absorption. Conversely, once the "lower" part is completely inserted in the snow, the absorption due to the rod becomes nearly constant while the signal continues to decrease as a result of the "normal" extinction caused by snow scattering and absorption. This explains why i) the log-radiance is lower when the rod is present compared to the ideal case (i.e. the ratio in Fig. 9b is less than 1), but ii) the decrease rates are almost equal under the transition zone near the surface.

To further investigate the impact of the rod, we run simulations with varying rod characteristics. Figure 10 shows the influence of rod diameter and albedo at $400\,\mathrm{nm}$. The departure from the ideal case increases with increasing diameter, probably because of the greater surface area available to intercept rays. However there is no clear linear or quadratic relationship with the diameter either in linear or in logarithm scales, suggesting a complex interaction. The light collected under the transition zone in depth is 1.5 and 2.2 times less intense with a $10\,\mathrm{mm}$ and $28\,\mathrm{mm}$-thick rod respectively compared to the ideal case. This is considerable if the data were to be directly interpreted in terms of actinic flux for photochemistry applications. Regarding the gradient estimation, it seems that the transition zone near the surface grows with increasing rod diameter. From a technical point of view, this suggests minimizing the size of the rod as much as possible. Likewise, embedding large spectrometers in snow to study extinction in the visible and UV is not recommended (Järvinen and Leppäranta, 2013). The albedo of the rod also has a great influence. Darker rods tend to absorb more and increase the artifact in terms of absorption and extent of the transition zone. At large depths, the light collected is half as intense with a 10-mm diameter rod albedo of 0.6 compared to the ideal cases while it is 1.5 with our rod with an albedo of 0.9. Future improvement of the system should seek more reflective paint or materials.

The rod itself is not the only component perturbing the profiles. To perform the hole before the insertion of the rod, we use a metal stick similar to the rod and that uses the same guide as the fiber. Despite this precaution, the alignment is rarely perfect which enlarges the hole, letting a small void between the snow and the rod. The light can propagate without loss in this void, which is illustrated in Figure 11. In contrast to the direct effect of the rod, the log-radiance gradient is affected at any depth. This is due to the light coming from brighter layers located above and propagating downward without extinction. Overall this tends to reduce the vertical gradient, as much as the hole gets bigger. The effect is larger at longer wavelengths (not shown) because the extinction is larger whereas the propagation in the void is wavelength-independent. The artifact due to the hole could be serious for the estimation of the ice absorption. Nevertheless, the simulations indicate that only large holes have a significant impact (typically 30-mm diameter and beyond). To our experience, void space less than $2\,\mathrm{mm}$ around the rod is typical. Moreover, it is easy to see problems during the perforation and discard the hole accordingly. Note also that we systematically add a few millimeters of snow on the surface around the rod to cover the void space from direct sun beam.

The rod absorption also depends on the snow properties. Figure 12 shows the ratio between the simulations with and without rod at depth ($50\,\mathrm{cm}$) as a function of SSA and density. It shows a clear decrease of the ratio (i.e. increase of the rod absorption) with both density and SSA. The sensitivity seems larger to the density than to SSA (when considering relative variations of these variables) and the relationship is not linear (isolines are not straight lines and the space between them is not constant).

This suggests a relationship involving the length scale $1/k_e$ which is inversely proportional to $\rho\sqrt{\text{SSA}}$ but further exploration of this hypothesis is let to future work. The best conditions to perform measurements is light snow with low SSA which is somewhat incompatible in practice as metamorphism usually tends to decrease SSA but increase density.

The first series of simulations shows that the measured radiance is significantly lower than in the ideal case and that the decrease rate is negatively biased with respect to the theoretical AFEC, but under the transition zone, this bias becomes small compared to the snow extinction. It implies that to retrieve ice absorption, the top $\approx$7–10 cm of the snowpack must be discarded. However, these simulations only consider an homogeneous snowpack which is a great simplification. Moreover, the dependency of the rod absorption to the snow properties highlighted in this section suggests that different results could be obtained in the case of real snowpack.

### 3.3.2 Real snowpack

The snowpack 25kmE_1 is now investigated. The MCML simulations are run with layers every 2.5 cm using measured SSA and density (Fig. 13, see also Libois et al., 2014a). The measurements and simulation results are shown in Fig. 14 at selected wavelengths. All the profiles are normalized by the radiance at 1 cm depth. IA2008 is used for the ice absorption in all the simulations.

The simulations at 700 and 600 nm with MCML show a very good agreement with SOLEXS observations up to 20 cm depth, and only slightly degrade below. It means that SSA and density measurements as well as the values of the constants $B$ and $g$ are accurate enough. The simulation which does not include the rod tends to over-estimate the radiance compared to the observations. This result clearly demonstrates that MCML captures the effect of our rod. At 500 nm, the agreement is not as good especially in the upper part (0–5 cm) and under 20 cm depth. At 400 nm, the model largely over-estimates the observed radiance at any depths. In addition, the AFEC is different which indicates the problem is not related to the rod or the model but the ice absorption (SSA or density error are also excluded because of the agreement at longer wavelengths). We ran the same simulations except IA2008 was replaced by the BAY ice absorption from Section 3.2 at 400 and 500 nm. The results show a clear improvement (Fig. 15), demonstrating that the ice absorption in IA2008 is too low to explain our observations. The slight degradation from IA2008 to BAY at 500 nm can be understood by noting that the absorption estimated on all the sites is slightly larger than the estimate based only on the 25kmSEO profiles (Fig 7).

Figures 14 and 15 show another interesting feature relevant to the estimation of the AFEC. The gradient changes not only near the surface but also around 15 and 20 cm. This is visible both in the observations and in the simulations either with or without the rod. This change coincides with the transition between two different layers noticeable in the field and visible in the SSA and density profiles (Fig. 13). However the change of gradient is more marked, nearly reaching the vertical at 400 nm, when the rod is present. Positive gradients can even be observed in some profiles (Supplementary Figure 1) which is theoretically impossible for a layered medium according to the radiative transfer theory. This result can be explained by the change of snow properties between 15 and 25 cm depth (Fig. 13) and its influence on the rod absorption (Fig. 12 where trajectory of SSA, density in the snowpack 25kmE_1 is reproduced in green). The peaks of SSA and density around 15 cm depth are favorable to a large absorption that appears as a low ratio in Fig. 14. Below, both SSA and density decrease and so

the ratio increases, which nearly compensates for the "normal" decrease expected from the snow extinction and results in a nearly null gradient. The issue is particularly complex because the rod absorption at a given depth is not only related to the SSA and density at the same depth. The comparison of profiles indeed shows that although the SSA and density transition is sharp around 20 cm, the transition from high to low rod absorption is more diffuse and seems to be located slightly lower. This

offset can be understood by the fact that the rod absorption is significant only over a few centimeters above the fiber tip, what was referred to as the "lower" part of the rod in Section 3.3.1. Hence, the gradient of the log-radiance is affected by the rod absorption as long as this lower part of the rod is in a transition between two different homogeneous layers.

A practical consequence of this issue is that even in homogeneous snow layers, the gradient over a few centimeters in the upper part of this layer is affected by the rod absorption in the overlying layer. This upper part must be excluded from the

estimation of the AFEC. Fortunately, this is what happens in practice because the selection of the homogeneous zones is done directly on the radiance profiles. We can expect that selecting linear pieces of profiles remove the effect of sharp transitions. Conversely, if the snow properties vary gradually with depth, the variations of the rod absorption might be smooth and be misinterpreted as linear piece. For instance, if the rod absorption is decreasing (e.g. because SSA is smoothly increasing), the gradient under-estimates the AFEC of the ideal case, resulting in an under-estimation of the ice absorption. Considering that at

Dome C the density is highly variable while the SSA tends to decrease in the first upper meter of the snowpack (Libois et al., 2014a), we expect that on average the rod absorption leads more frequently to under-estimation of the ice absorption. This results also in the counter-intuitive conclusion: a more absorbing rod (i.e. either because it is larger or darker) can yield a lower ice absorption estimation in some cases. It is worth mentioning that the problem of the rod absorption is evaluated here with a specific objective in mind, but it concerns any study exploiting measurements with an inserted optical fiber (e.g. King and

Simpson, 2001; France and King, 2012). The interpretation of such measurements remains however highly tied to the specific protocol used. The next section quantifies the ice absorption uncertainty due to the impact of the rod.

### 3.3.3   Impact of the rod on the ice absorption estimation

The uncertainty range caused by the rod-snow interactions is evaluated here by considering the two snowpacks investigated earlier: the homogeneous snowpack which leads to an over-estimation of the AFEC and the 25kmE_1 snowpit which results

in the opposite. To perform this evaluation, we ran MCML for each snowpack and for each ice absorption spectrum (IA2008 and BAY) which yields simulated radiance profiles as in Figures 9, 14, 15. We then apply the WGB method on these simulated profiles exactly as if they had been measured with SOLEXS. This yields the absorption spectra plotted in Figure 16. They ideally should be equal to the ice absorption spectra used as input for the simulations but differ because of the rod absorption and the properties of the snowpack.

Figure 16 shows as expected that the homogeneous snowpack results in an over-estimation of the ice absorption and the opposite is true for the 25kmE_1 snowpack. The range between these two snowpacks is larger (1 order of magnitude) with the lower absorption spectrum (IA2008) than with the BAY estimate (a factor of 2 in linear scale) in the range 350–550nm. In both cases, this uncertainty is larger than the statistical uncertainty estimated from the posterior (Section 3.2) which indicates that the rod absorption dominates the error budget. Nevertheless, the uncertainty ranges around IA2008 and BAY do not overlap

which means that the rod absorption effect is insufficient to explain the discrepancy between both ice absorption estimates, at least considering that Warren et al. (2006) rod and snow properties were relatively similar to ours.

## 4  Discussion

The ice absorption obtained with our new observations is one order of magnitude greater than the most recent and widely-used compilation proposed by Warren and Brandt (2008) at 400–450 nm. Although, we have not found a satisfying explanation for this discrepancy, we show here some potential artifacts (instrumental and methodological artifacts, environmental conditions, and contamination by light absorbing impurities) and argue that they can not quantitatively explain the discrepancy.

First, the large difference has been obtained whatever the estimation method and filtering of the data which demonstrates that the difference is statistically significant and robust against the methodology. The simulations with MCML showed that the instrument is not neutral, resulting in differences between the vertical gradient of the measured log-radiance and the asymptotic flux extinction coefficient $k_e$, jeopardizing the estimation of the ice absorption. The conditions in which this difference is small were determined based on the simulations. We found that the linearity of the log-radiance profile and distance from the surface are the two main criteria which were then used to determine by visual selection the homogeneous layers in the profiles. This step is in some way subjective and is imperfect because the linearity of the log-radiance profiles is not strictly synonym of homogeneity of the snow properties (SSA and density). Part of the scatter in the absorption estimated from each layer (Fig. 5) could be explained by selection errors, but we exclude that a bias as large as the difference between our estimate and IA2008 would result from this step. We have shown that with our rod characteristics the impact is moderate in the case of high absorption (around $10^{-2}\,\mathrm{m}^{-1}$). Since the approach followed by Warren et al. (2006) is also based on the selection of homogeneous zones and only two zones were considered, it cannot be excluded that layer C was a statistical outsider. Nevertheless, this hypothesis has a low probability because none of our 70 homogeneous zones yields absorption spectra as low as IA2008. Other causes including possible heterogeneity (trend in SSA or density), the hole around the rod (Fig. 11), illumination variations and other details of the protocol that we are not aware of are more likely but none can be easily identified. All in all we believe the BAY estimate is not less reliable than other estimates.

Second, despite many similitudes between our experimental protocols and that of ?, the environmental conditions were slightly different. Figure 17 reports again our estimates and IA2008 along with Antarctic Muon and Neutrino Detector Array (AMANDA,  Askebjer et al., 1997; Ackermann et al., 2006) estimates. The latter spectra are intermediate between IA2008 and our estimate. This follows the temperature since IA2008 measurements were taken between 1 and 2 m depth where the temperature is around -55°C, our measurements comes from near the surface where the temperature is higher in summer (around -30°C) and AMANDA measurements were taken much deeper in the ice-sheet where the temperature is intermediate. However, Woschnagg and Price (2001) indicate that the ice absorption is insensitive to pressure and the sensitivity to temperature is only of the order of $+1\%\,\mathrm{K}^{-1}$ which is largely insufficient to explain a differences.

Third, even if the absorption coefficient is accurately estimated, the question of the contamination of the snow remains. Figure 17 shows calculations of the equivalent absorption coefficient of snow containing small amount of black carbon using

IA2008. The latter calculations consider the homogeneous snowpack studied in Section 3.3.1 and use the TARTES model following Libois et al. (2013) setup with IA2008 ice absorption. Soot properties (density, absorption spectra) are still uncertain (Bond and Bergstrom, 2006) so that the concentrations used for these simulations should be considered as approximate. The soot particles are supposed to be located outside the ice grains, which tends to lower the efficiency of its absorption compared

to internal mixture (Flanner et al., 2012). TARTES simulation using IA2008 shows the discrepancy between IA2008 and BAY is equivalent to about $5\,\mathrm{ng\,g^{-1}}$ of BC (Fig. 17). However, the amount of light absorbing impurities (LAI) is usually lower than that in Antarctica but the range of reported measurements and estimates is large. A recent review by Bisiaux et al. (2012) summarized many previous measurements of black carbon (BC). BC concentration is often under $0.3\,\mathrm{ng\,g^{-1}}$ in remote continental Antarctic locations, except in the vicinity of research stations. For instance, Warren and Clarke (1990) measured

only $0.1$–$0.3\,\mathrm{ng\,g^{-1}}$ of elemental carbon 13 km upwind from the station at South Pole but this value reached $3\,\mathrm{ng\,g^{-1}}$ 1 km downwind. Grenfell et al. (1994) investigated in detail the spatial distribution of soot around Vostok station and revealed a clear pattern with maximum concentration around $7\,\mathrm{ng\,g^{-1}}$ in the downwind direction. At 6 and 10 km upwind, the concentrations were lower than $1\,\mathrm{ng\,g^{-1}}$ which is in agreement with South Pole upwind concentration once the dilution due to the 2–3-fold lower annual accumulation is taken into account. At Dome C, the shallowest layer in Warren et al. (2006) (Layer B) was

found to be contaminated ($1.2\,\mathrm{ng\,g^{-1}}$) due to recent human activities. This is the reason why the deepest layer (Layer C) was selected for IA2008. Zatko et al. (2013) measured $0.6\pm0.2\,\mathrm{ng\,g^{-1}}$ of BC on samples collected at 11 km from Dome C station in 2004. This "background" concentration is compatible with those measured at Vostok and South Pole taking into account the intermediate annual accumulation at Dome C. Even if the opening of the permanent station Concordia in 2005 has certainly intensified the emission and deposition, we believe that the Vostok study (Grenfell et al., 1994) provides a suitable reference

for a permanent station. It is therefore unlikely that significant BC contamination could be found beyond a few kilometers upwind. Although BC is a major LAI, it is not the only one. Royer et al. (1983) measured dust concentrations of $26\,\mathrm{ng\,g^{-1}}$ at Dome C, which is optically equivalent to less than $0.15\,\mathrm{ng\,g^{-1}}$ of BC (Warren and Clarke, 1990). By measuring the absorption of residuals in filtered samples following (Grenfell et al., 2011), Zatko et al. (2013) determined the contribution of the non-BC LAI to total absorption in the surroundings of Dome C. They found a contribution of 30% in the range 650-700 nm and of 75%

at 400 nm. Equivalent BC concentrations in the UV could thus be much larger than actual BC concentrations, but their impact should remain moderate in the visible part of the spectrum on which the present study focused. This short review suggests that contamination level as high as those required to explain the difference between IA2008 and our estimates ($\approx 5\,\mathrm{ng\,g^{-1}}$ in Fig. 17) can only be found downwind close to the stations. It is therefore very unlikely that our upwind sites at 25 or even 10 km from Concordia would reach such high equivalent BC concentrations. The apparent dependence of the absorption to the

distance (and downwind direction) shown in Figure 8 is not understood, but even if it was attributed to contamination, it would indicate the uncertainty due to impurities is small (at most a factor of 2) compared to the differences between the different estimates.

   Despite no explanation of the discrepancy can be given, a convergence of facts tends to suggest that the ice absorption IA2008 is too low: i) this value was associated with a large uncertainty (non-monotonicity of the slope in figure 7 of Warren

et al. (2006)); ii) such low estimate has never been confirmed neither by using a similar instrument and method (this study), nor

by other independent methods (Askebjer et al., 1997; Ackermann et al., 2006); iii) the difference between Layers B and C of Warren et al. (2006) could not be conciliated with the BC concentration measured in Layer B, highlighting some inconsistency in either one or both of these estimates; iv) the simulation with MCML have shown that large under-estimations of the ice absorption caused by the rod-snow interaction are possible; at last v) AMANDA estimates are relatively closer to ours even

if differences persist. This convergence would have been even better if we had chosen one of the AMANDA estimates as the reference at 600 nm (and longer wavelengths) instead of values from IA2008. Therefore, we propose to update IA2008 with a higher absorption value in the visible and UV. For this, and to reduce the risk of including potentially contaminated sites, we computed another estimate using only the sites at 6 km or more from Concordia station and by excluding the downwind, North, direction. This new estimate, called 'clean', is based on 46 homogeneous zones which is still large compared to previous

studies. It appears to be very close to the one including all sites (Fig. 17), which highlights that this selection is only a precaution without significant quantitative impact. Supplementary table 1 provides mean and standard deviation of the samples drawn from the posterior and data are available from http://lgge.osug.fr/~picard/ice_absorption/. More importantly than adopting an estimate or another, we recommend modelers to systematically account for the large uncertainty on ice absorption that persists even after the present study.

To briefly illustrate the potential impact of our new estimate, we computed the change of albedo between IA2008 and BAY for a homogeneous snowpack with SSA=30 $m^2kg^{-1}$ using TARTES. The illumination is direct with $53°$ zenith angle. The difference is the largest in the UV, around 0.008 and decreases almost linearly to zero at 600 nm (not shown). Even though this value seems weak, it is not negligible because solar irradiance is maximum at the shorter visible wavelengths. The total irradiance between 350 and 600 nm measured at Dome C on 10 January 2013 at local noon (figure 7 in Picard et al., 2016)

amounted to 254 $W\,m^{-2}$. Combined with the albedo calculation, the energy absorbed by the snowpack is 2.0 $W\,m^{-2}$ with IA2008 and 2.9 $W\,m^{-2}$ with the 'clean' estimate, a difference of 0.9 $W\,m^{-2}$. This difference is small but not negligible at climate time scales. A much greater impact is expected on the penetration depth which is relevant to for snow photo-chemistry. A calculation on the same homogeneous snowpack shows that the actinic flux at 25 cm depth at 400 nm is nearly 3 times lower using our estimate than using IA2008.

**5 Conclusions**

This study reproduced the method developed by Warren et al. (2006) to estimate the ice absorption coefficient in the visible. Using radiance measurements in 70 homogeneous layers and Bayesian inference, it provides a new estimate of the ice absorption coefficient in the visible. This estimate is around $10^{-2}\,m^{-1}$ at the minimum absorption, which is significantly larger than the Layer C estimate obtained by Warren et al. (2006) and widely used as a reference (Warren and Brandt, 2008) in the range

390–600 nm. This difference is equivalent to contamination of snow by 5 $ng\,g^{-1}$ of externally-mixed BC which is much larger than BC concentrations observed a few kilometers away from research stations on the Antarctic Plateau. The same conclusion applies for dust after taking into account the equivalent radiative effect. Our estimate of ice absorption is closer to other es-

timates which are based on different measurement techniques (Ackermann et al., 2006). Overall, the differences have not be explained in the present, and our main conclusion is that ice absorption remains largely uncertain in the range 350–500 nm.

Considering the lack of reproducibility between Warren et al. (2006) and the present study, the potential artifacts due to the instrument in the case of low absorption (Section 3.3) and the relative convergence of other estimates, we however suggest that the convergent values from AMANDA or this study should be used in the visible and UV. Future work should aim at improving the measurement technique and the method by exploiting the 3D radiative transfer simulations to design less intrusive instruments and to explicitly account for the snow-instrument interaction during the ice absorption estimation process. Conducting new measurements further from the stations and where the snowpack is more homogeneous than at Dome C is also an avenue.

*Acknowledgements.* This study was supported by the ANR program 1-JS56-005-01 MONISNOW. The authors acknowledge the French Polar Institute (IPEV) for the financial and logistic support at Concordia station in Antarctica through the NIVO program. The computations have been done on the CIMENT cluster. The authors thank S.G. Warren and R. E. Brandt for discussions about their experimental setup and their comments on an earlier version of the manuscript. The Editor and two anonymous Reviewers are acknowledge for their suggestions to improve the manuscript, especially regarding the differences between the various ice absorption estimates. M. Dumont is also acknowledged for her comments on the initial manuscript.

*Author contributions.* L. Arnaud and G. Picard developed SOLEXS. G. Picard developed the 3D Monte-Carlo Radiative code. G. Picard and Q. Libois developed the Bayesian inference method. All the authors contributed to the measurements, data analysis and preparation of the manuscript.

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

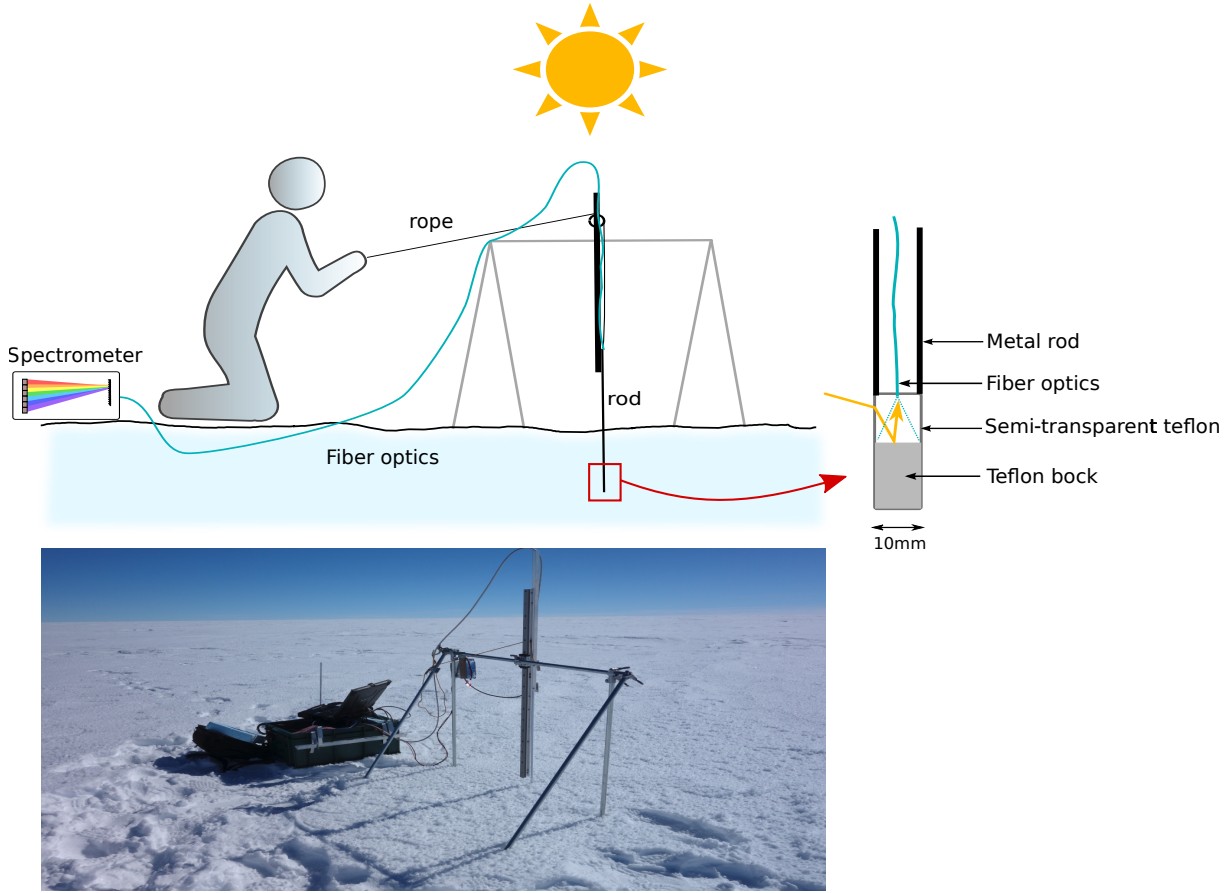

**Figure 1.** Scheme and picture of SOLEXS. The rod (black) is vertically guided and inserted in the snow with the rope manipulated by the operator. The quadripod (gray) is oriented to minimize the shadows. At the bottom end of the rod, the light first traverses the semi-transparent 0.5-mm thin teflon cylinder and is reflected by a teflon block towards the fiber tip from where it is transferred to the spectrometer.

Wiscombe, W. J. and Warren, S. G.: A Model for the Spectral Albedo of Snow. I: Pure Snow, J. Atmos. Sci., 37, 2712–2733, doi:10.1175/1520-0469(1980)037, 1980.

Woschnagg, K. and Price, P. B.: Temperature dependence of absorption in ice at 532 nm, Applied Optics, 40, 2496, doi:10.1364/ao.40.002496, http://dx.doi.org/10.1364/AO.40.002496, 2001.

5  Zatko, M. C., Grenfell, T. C., Alexander, B., Doherty, S. J., Thomas, J. L., and Yang, X.: The influence of snow grain size and impurities on the vertical profiles of actinic flux and associated NOx emissions on the Antarctic and Greenland ice sheets, Atmospheric Chemistry and Physics, 13, 3547–3567, doi:10.5194/acp-13-3547-2013, http://dx.doi.org/10.5194/acp-13-3547-2013, 2013.

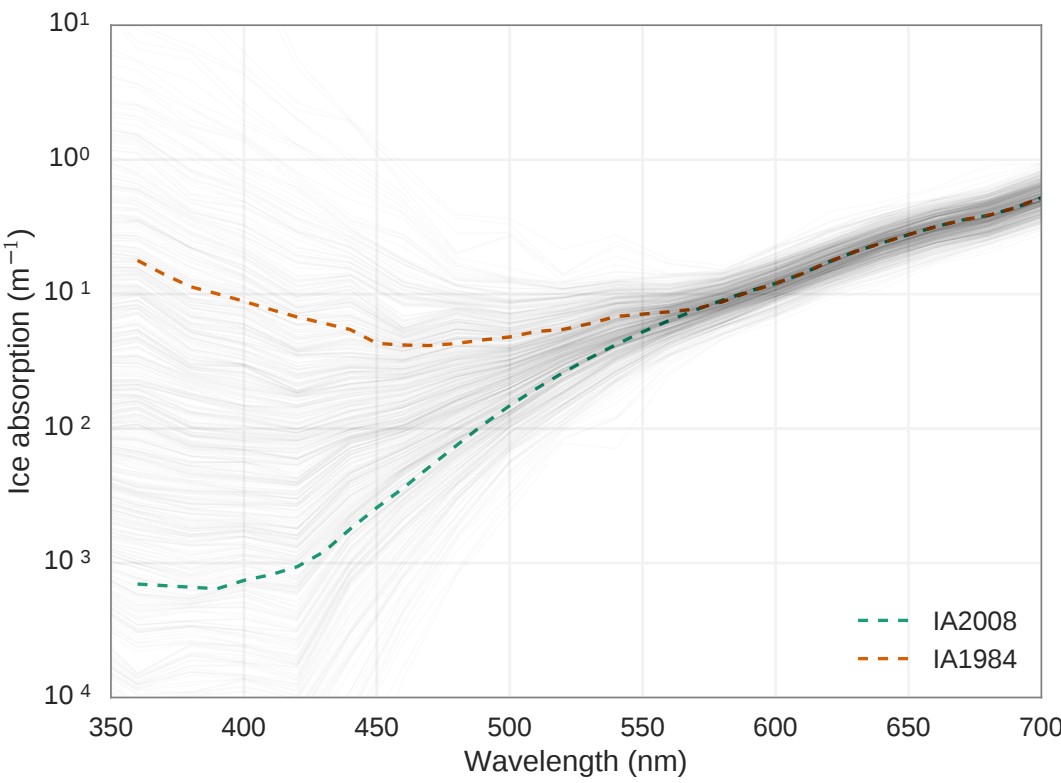

**Figure 2.** Samples (gray) of the prior distribution of the ice absorption to be used by the BAY method. Ice absorption coefficients IA1984 and IA2008 are presented for reference.

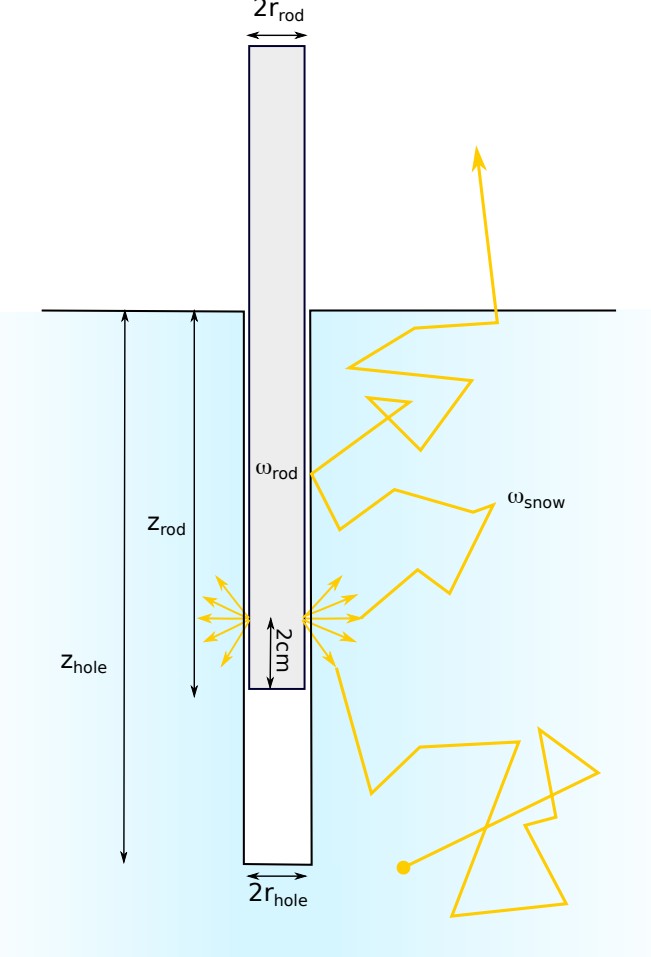

**Figure 3.** Principle of the adapted 3D radiative transfer model MCML. The rod (gray) is a cylinder of diameter $2r_{rod}$ inserted down to $z_{rod}$ depth in the hole. The hole is also a cylinder of diameter $2r_{hole}$ and extends up to $z_{hole}$ depth in the snow. Rays (orange) are launched from the sensor entrance which is located $2\,\mathrm{cm}$ above the end of the rod and backpropagated. They eventually reach the surface or are discarded when their energy is lower than a small portion of its initial value ($10^{-5}$).

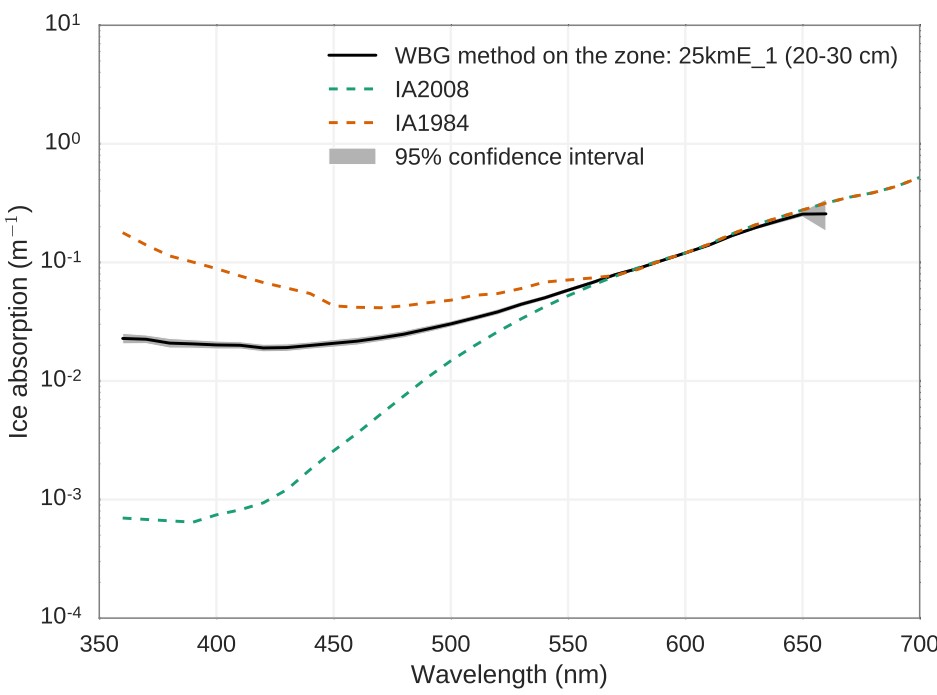

**Figure 4.** Ice absorption estimated by the WBG Method (black) on the SOLEXS profile at 25kmE_1 (25 km East of Concordia). The 95% confidence interval is shown in gray shade.

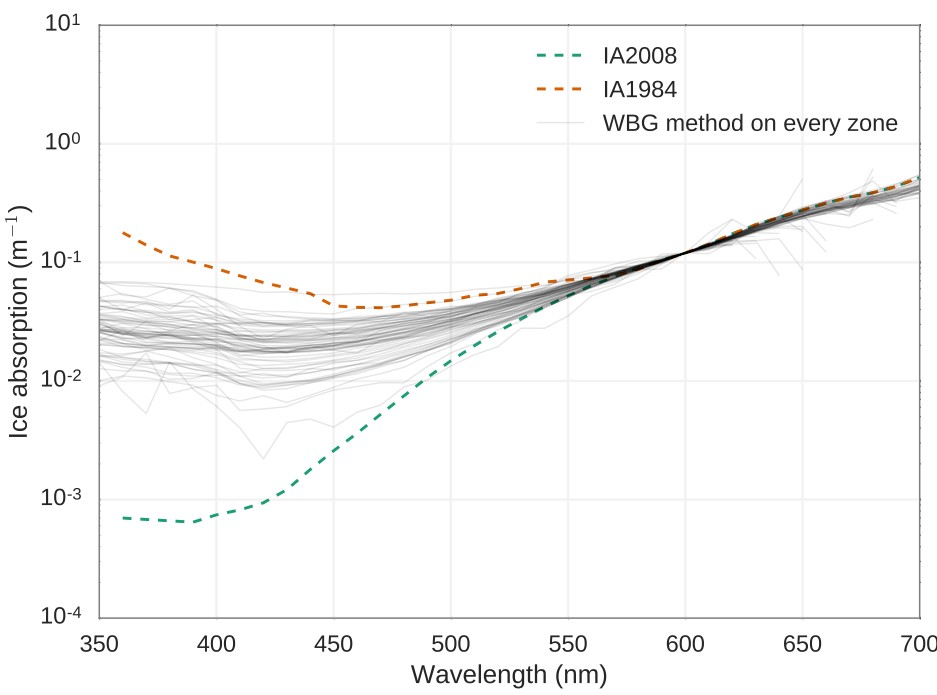

**Figure 5.** 70 ice absorption spectra estimated by the WBG method (Warren et al., 2006) on each homogeneous zone from the selection on the 56 radiance profiles measured around Concordia.

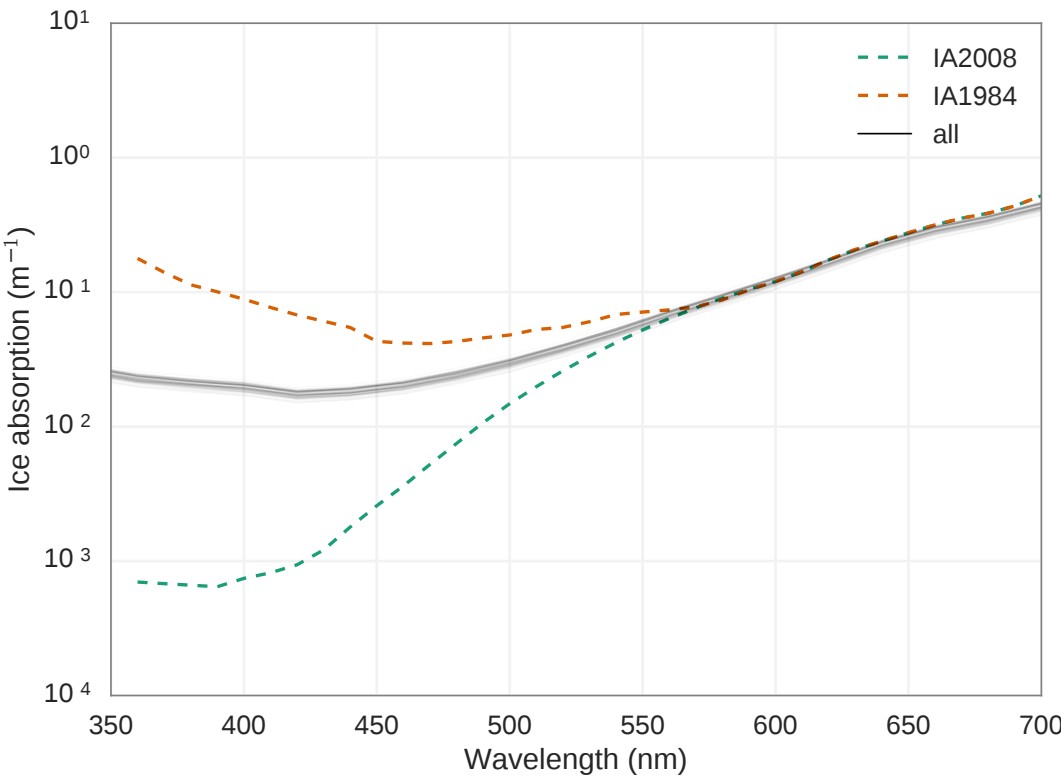

**Figure 6.** Samples (gray) of the posterior distribution of the ice absorption estimated by the BAY method on the 70 homogeneous zones selected in the 56 radiance profiles measured around Concordia.

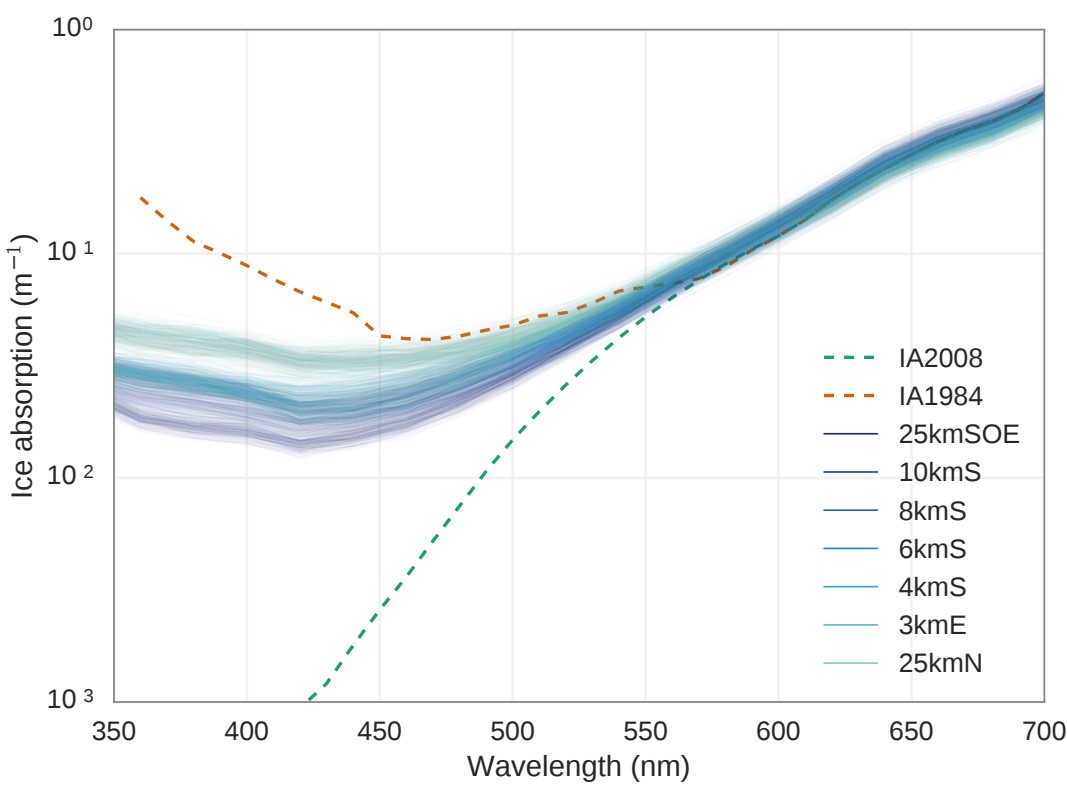

**Figure 7.** Ice absorption estimated by the BAY method at different locations around Dome C.

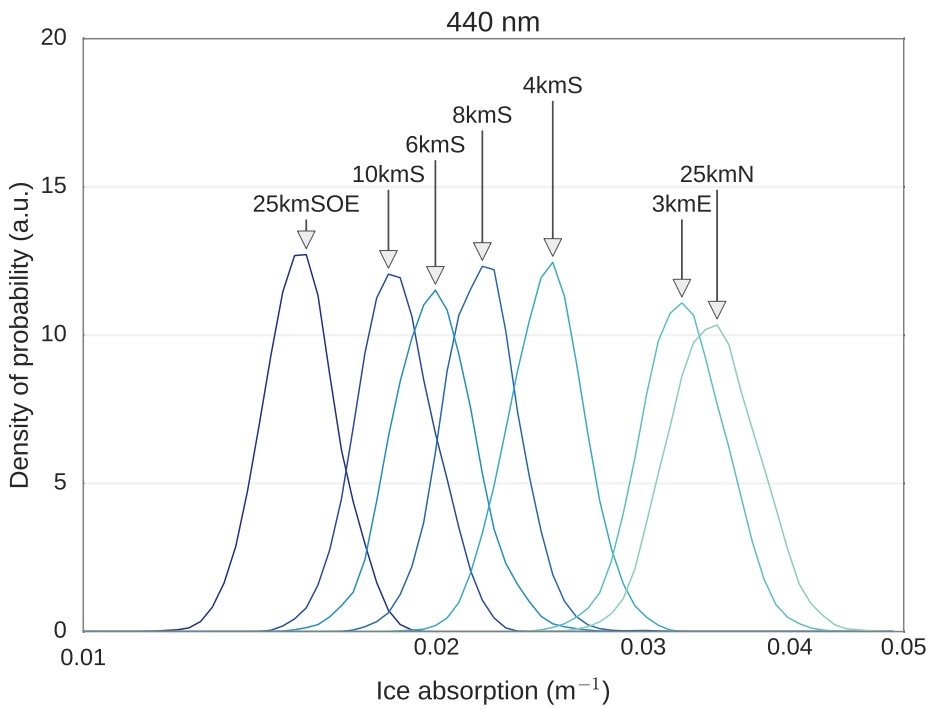

**Figure 8.** Posterior of ice absorption at 440 nm independently estimated on different sites by the BAY method.

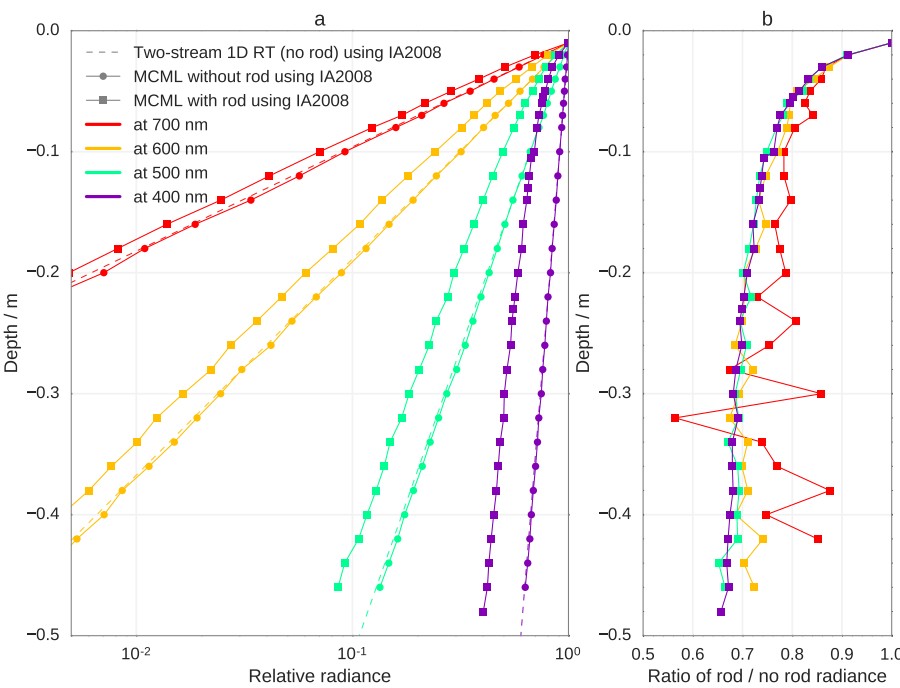

**Figure 9.** a) Log-radiance profiles at 400, 500, 600 and 700 nm simulated with the 1D model TARTES (without rod) and the 3D model MCML (with and without rod) for an homogeneous snowpack with SSA of $30\,\mathrm{m^2 kg^{-1}}$ and density of $350\,\mathrm{kg\,m^{-3}}$. b) same as a) but normalized by the ideal case computed with TARTES.

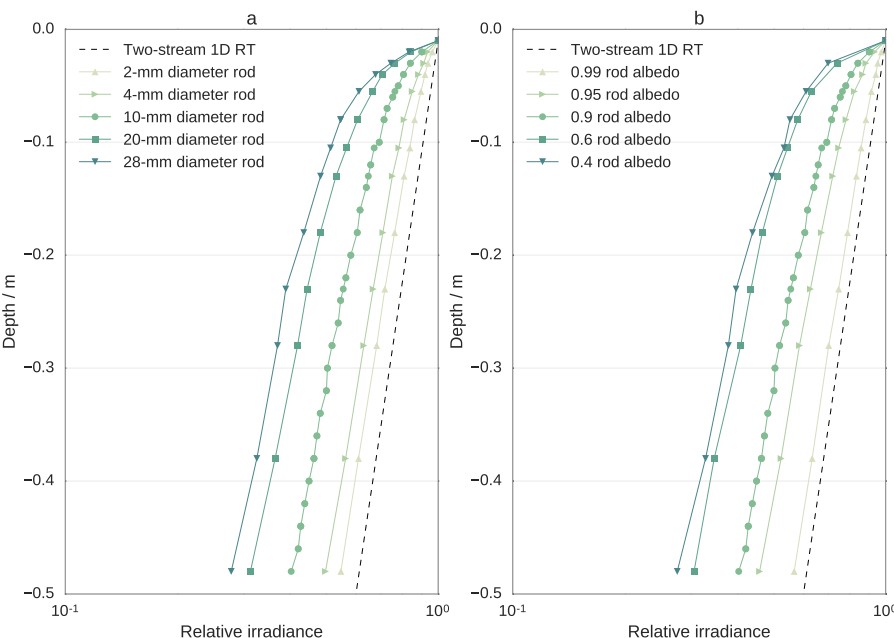

**Figure 10.** Influence of the a) rod diameter and b) rod albedo on the log-radiance profiles at 400 nm simulated by MCML for an homogeneous snowpack with SSA of $30\,\mathrm{m^2\,kg^{-1}}$ and density of $350\,\mathrm{kg\,m^{-3}}$. The 1D model shows the reference profile in the ideal case (no rod).

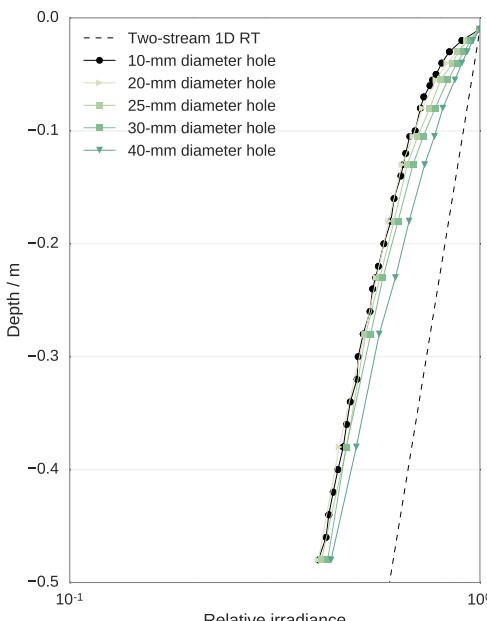

**Figure 11.** Influence of the air gap between the 10-mm diameter rod and snow simulated with MCML at $400\,\mathrm{nm}$ for an homogeneous snowpack with SSA of $30\,\mathrm{m^2\,kg^{-1}}$ and density of $350\,\mathrm{kg\,m^{-3}}$. The 1D model shows the reference profile in the ideal case (no rod and no gap).

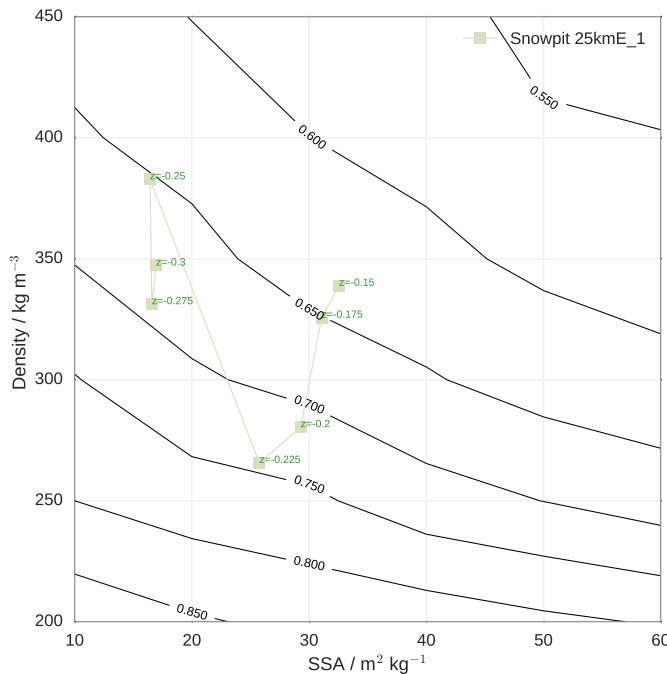

**Figure 12.** Ratio at 50 cm depth (black contour lines) between the radiances calculated by MCML simulations with rod and without rod for an homogeneous snowpack with varying SSA and density. Low values of the ratio indicate high rod absorptions. Green symbols represent the couples (SSA, density) at several depths measured in the 25kmE_1 snowpit.

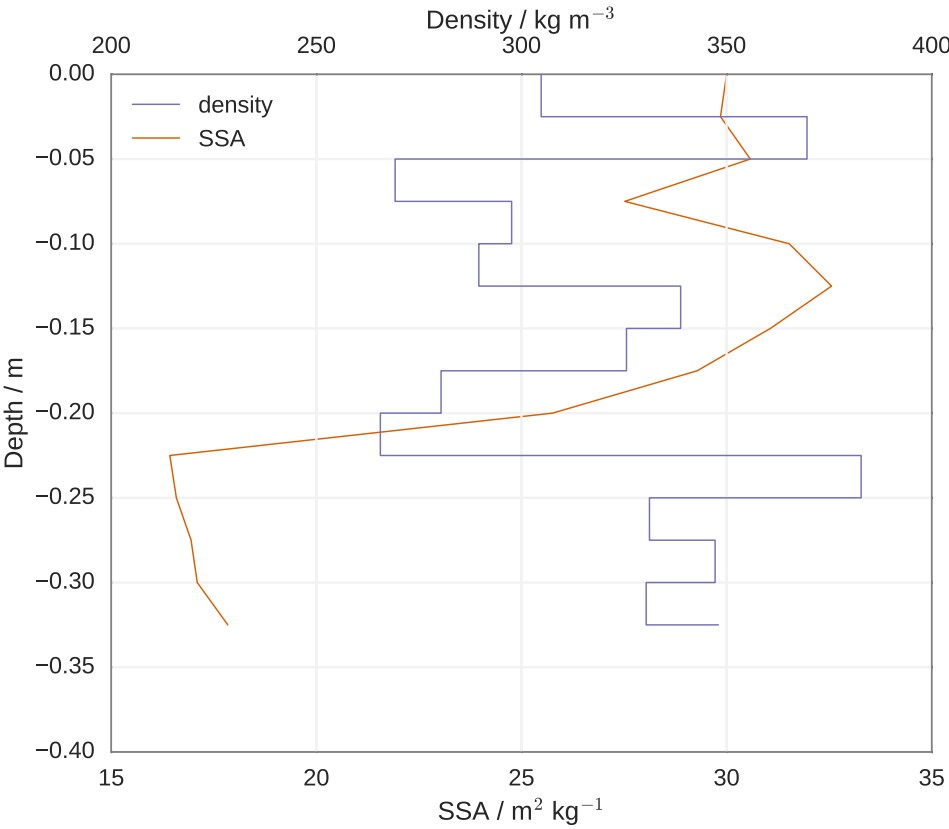

**Figure 13.** Vertical profiles of SSA (in orange) and density (in blue) measured in the 25kmE_1 snowpit.

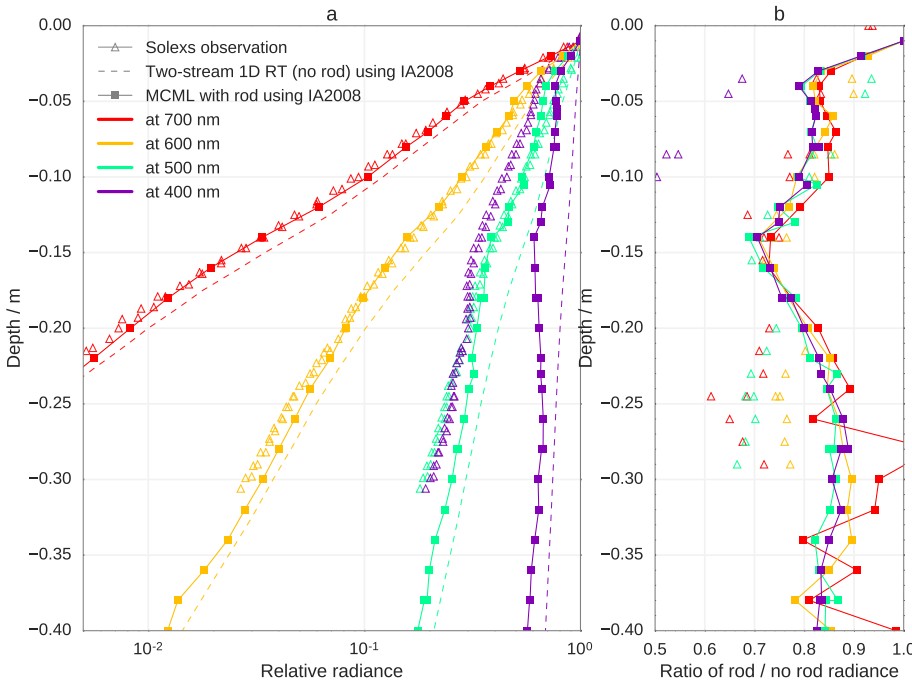

**Figure 14.** Log-radiance profiles at 400, 500, 600 and 700 nm measured by SOLEXS at 25kmE_1 (25 km East of Concordia). Simulations with the 1D model TARTES (without rod) and the 3D model MCML using density and SSA profiles measured at the same point (Libois et al., 2014b). b) same as a) but normalized by the ideal case computed with TARTES.

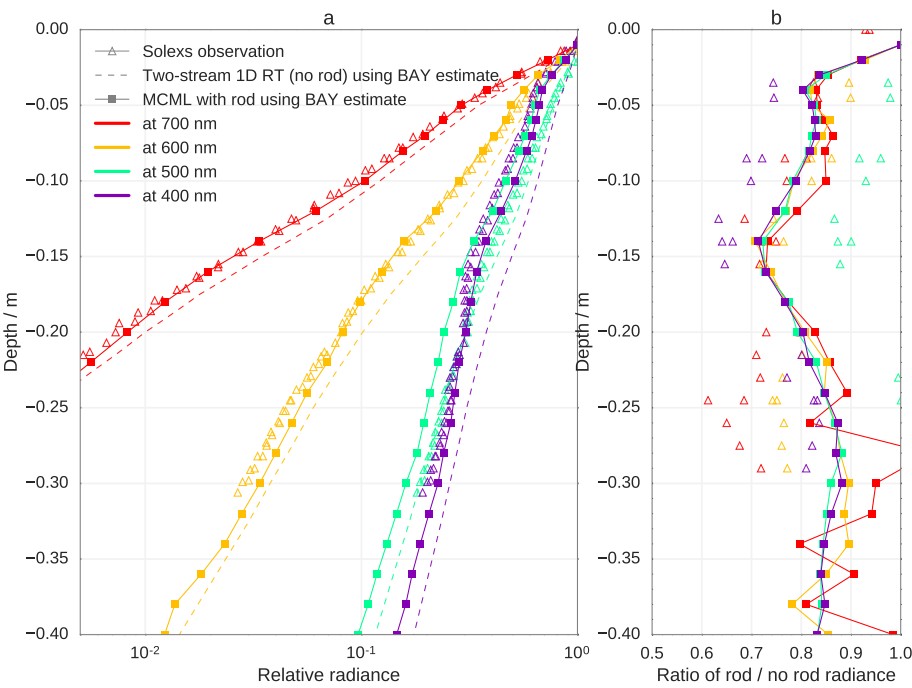

**Figure 15.** Same as Fig. 14 except that the simulations at 400 and 500 nm uses the BAY estimate of the ice absorption coefficient.

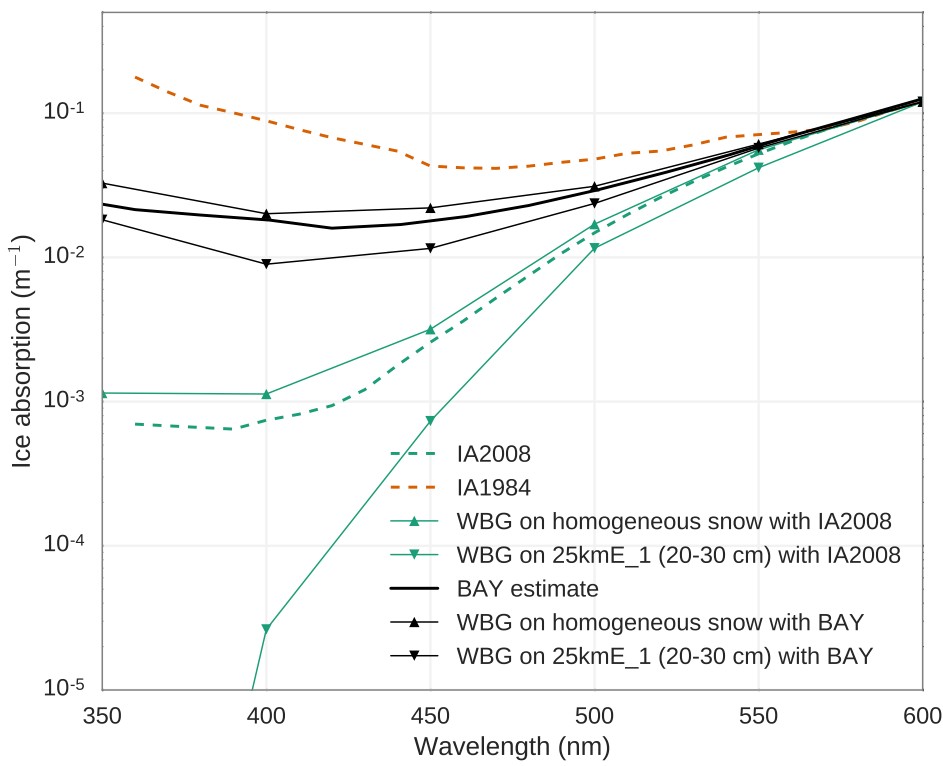

**Figure 16.** Ice absorption estimated with the WGB method applied to simulated irradiance profiles with MCML in the presence of the rod for the homogeneous snowpack (triangles up) and the 25kmE_1 snowpit (triangles down) by using IA2008 (green) and BAY (black) ice absorption spectra, respectively, as input.

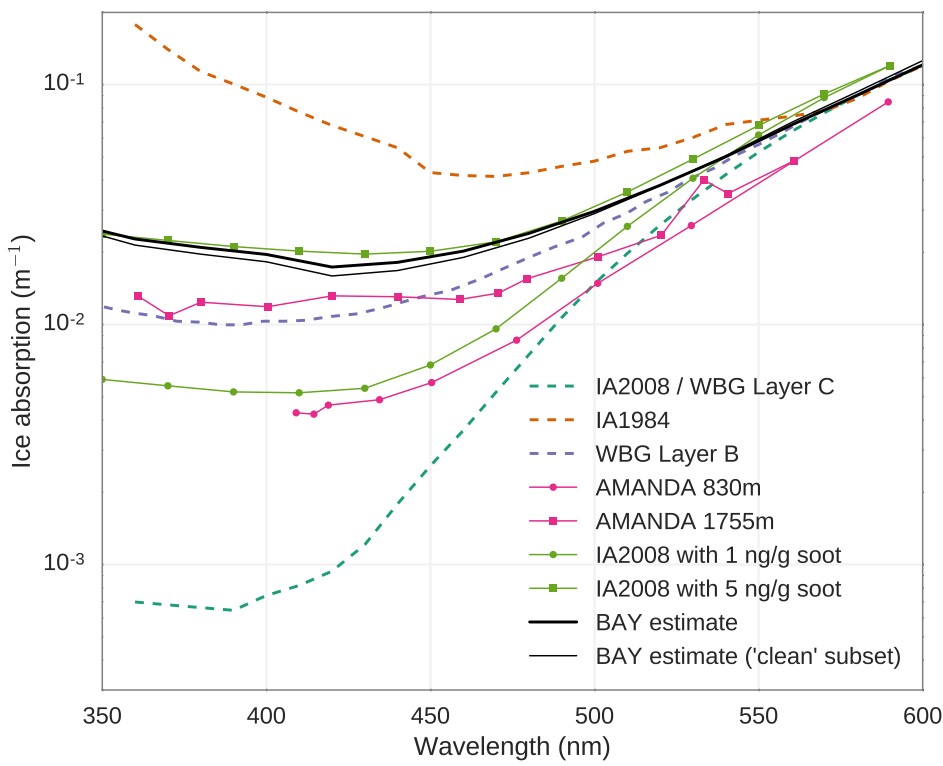

**Figure 17.** Estimates of ice absorption from this paper (black, BAY) from AMANDA experiment (Ackermann et al., 2006) (pink) and from Warren et al. (2006) (blue, green, maroon). Simulations (green) show absorption of snow contaminated with soot and considering IA2008 for pure ice.