# Peer review of "Refinement of the ice absorption spectrum in the visible using radiance profile measurements in Antarctic snow"

_The Cryosphere, 2016_

## Referee Comment (RC1) · Anonymous Referee #1 · 28 Jul 2016

This study presents new estimates of visible ice absorption coefficients, using a large number of measurements of light extinction in Antarctic snow. Precise estimates of the UV and visible absorption coefficient have eluded the scientific community because ice absorbs so weakly at these wavelengths, presenting numerous analytical challenges. This study builds on that of Warren and Brandt (2008) by (1) applying a larger set of measurements, (2) applying Bayesian statistical techniques that incorporate measurements at a larger number of wavelengths, and (3) conducting a rigorous modeling assessment of possible biases introduced by the presence of the fiber optic sensor and housing rod in snow. The current study finds that ice absorbs more strongly in the short-wavelength visible than found by Warren and Brandt (2008), though for unknown

reasons. Overall, this is an important and very thorough study, certainly worthy of publication. The assessment of potential sensor biases using 3-D Monte Carlo modeling seems particularly rigorous and informative, and I think this aspect of the study will help inform past and future measurements of radiance extinction with depth in snow. I also expect that the new estimates of ice absorption spectra, available as supplementary data to the paper, will be widely used in the scientific community. It would have been satisfying if the authors had presented a convincing reason (or set of reasons) for why their estimates differ from those of Warren and Brandt (2008), but such an assessment may not be possible, or is otherwise beyond the scope of this study. It seems possible or likely that a combination of factors contributed to these differences. Below are some minor issues for consideration. Overall, I think this is an excellent study.

Minor comments:

p1,6: "larger than IA2008 by one order of magnitude..." - Larger at 400 nm, or averaged over the spectra? Please clarify.

p3,1-2: The inference from the quotation from Warren et al (2006), suggesting that more measurements should be made in snow further from sources of contamination, is that IA2008 measurements could be biased towards being too absorptive, but in fact this paper shows the opposite. Although this quotation is presented merely for motivation, reasons for the different findings between these studies appear to remain unknown. In general it would be helpful to offer (elsewhere in the paper) any additional insight or speculation that you have on reasons for the differences between these two studies. Convenient places for such discussion include sections 3.1, 3.4, and section 4.

Equation 1: Technically, "I(z=0)" should be "I(z=0,$\lambda$)" for consistency with the left-hand side of the equation.

p5,20: Please clarify what is meant by "the first two factors on the right hand side". The square root term makes this statement a bit ambiguous.
p5,19 and equation (3): It appears that the $\sigma$ terms refer to extinction coefficients of ice+air. If so, please explicitly clarify this. Otherwise, the distinction between $\sigma_a$ and $\gamma_{ice}$ is unclear, as both are absorption coefficients with identical units.

Equation (3): It would be helpful to briefly explain the conditions under which the middle expression of Equation 3 apply, as it appears to be an approximation.

p5,26: Does parameter $B$ have much spectral dependence?

Equation (6): Please define the symbol $\alpha_n$ (perhaps accomplished most conveniently in the description of Equation 4).

p6,17: "$\sigma$ measures the observation errors which are assumed identical for all the measurements." - Are there any conceivable or plausible conditions where the observation errors would depend on one or more key variables, such as depth in snow, surface irradiance, SSA, snow density, etc? In other words, what is the validity of this assumption?

p6,34: "The most likely value is taken as the average of these two parameterizations" - Do you mean that the available supplementary data are taken as the simple average of the WBG and BAY techniques, as applied to measurements collected from this study? Please clarify.

p7,28: Presumably, the distribution of these step lengths is such that the extinction transmittance obeys Beer's Law. It could be worth mentioning nonetheless.

p7,31: Briefly, how is bias avoided? Is it simply because the cutoff threshold is sufficiently small?

p8,15: Mode 2 (inverse tracking) is a creative solution to this problem.

Section 3.3: This is a very informative and interesting analysis. It could also be very useful if you can offer any insight into how previous measurements of light extinction in snow should be re-interpreted, in light of this analysis.

12,31-32: This last sentence is unclear to me.

[Figure]

Section 3.5 and Figure 16: This analysis is unclear to me. Please consider revising the description of this sensitivity study for clarity. In particular, the determination of ice absorption using different ice absorption data (caption of Fig 16) seems circular. The third sentence of this paragraph (section 3.5) appears to explain the technique, but I don't quite understand what is being done here.

13,28-30: This sentence should be fixed for clarity.

15,17-18: Could the large underestimations of ice absorption caused by rod-snow interactions be sufficient to explain the discrepancies in absorption between this study and Warren and Brandt (2008)? Insight on this would be helpful.

Figure 6 caption: What is the site of these measurements? Is it the same as in Fig 5?

Figure 7: It is a bit difficult to match the colors of the lines to the legend, because they are so similar, but perhaps this is not important.

Figure 13: Needs a legend or color description in the caption to distinguish the two lines.

---

## Referee Comment (RC2) · Anonymous Referee #2 · 11 Aug 2016

This paper presents updated estimates of the optical ice absorption spectrum in the visible based on a large set of new measurements of light extinction near the surface in Antarctic snow. The very weak optical ice absorption in the UV and visible creates experimental challenges which different techniques have been used to overcome. Lab measurements have been of limited use due to the required long path lengths and very pure ice samples. Measurements in deep natural ice in Antarctica by AMANDA (e.g., Ackermann et al., 2006) found that absorption in the visible is still dominated by dust contamination even for this extremely clean ice so intrinsic absorption by pure ice is experimentally hard to disentangle from absorption by impurities. Therefore, it has not been possible to definitely determine how strong optical absorption by pure ice actually

is and existing measurements can in some sense only set upper limits, unless it can be shown that impurities and instrumental effects are negligible. Warren and Brandt (2008), referenced as IA2008, developed a technique to measure absorption through radiance measurements in highly-scattering Antarctic snow. Their measurement was in the end limited to a single snow layer, but they found an even weaker absorption than measured in deep ice by AMANDA. The current paper uses a similar technique as Warren and Brandt but improves on important aspects: a much larger data set, collected at different locations, is used; instrumental effects related to inserting the optical fiber assembly into the snow is studied with detailed 3D simulations; and they also use more sophisticated Bayesian statistical techniques to fit absorption parameters using all data at once. This work is very valuable in that it adds more information to the question of optical ice absorption near the minimum and also sheds more light on possible systematic uncertainties involved with such snow measurements. The paper should definitely be published, but I have some substantial comments on the current version.

Major comments:

1) The experimental technique, data collection, analysis, and bias discussion is thorough and described overall in a clear way (with some exceptions discussed below). My main comment concerns the interpretation of the result. Is the claim that these measurements arrive at an estimate of the absorption coefficient for pure ice, i.e. the intrinsic absorption by ice without impurities? It would be hard to make this case, considering previous measurements. We know from AMANDA that ice absorption which is still dominated by dust contamination is weaker than these new results. Therefore, the even weaker absorption in IA2008 logically comes closer to the true absorption for pure ice. The weakest absorption measured in AMANDA dips below $5 \times 10^{-3}$ m$^{-1}$, but this is still in ice with considerable dust and the spectral shape is the power law expected from absorption by dust. I would therefore expect pure ice absorption in the visible to be even weaker. The new BAY (clean) measurements show much stronger absorption than the cleanest AMANDA depths. We know that pure ice is at most as absorbing as

the AMANDA ice. So the BAY estimate seems too absorbing. In this context it would also make sense to soften the definitive statement (Page13Line22) that "it is impossible to obtain absorption coefficient as low as IA2008". The large difference between the very weak IA2008 absorption and these new measurements with a similar technique is not understood, but logically IA2008 should be closer to true ice absorption since the AMANDA measurements set an upper limit.

2) I would have liked to see the plotted data and simulation results shown with estimated uncertainties (error bars and bands) whenever possible to aid the interpretation. Can the statistical and systematic uncertainties of the absorption spectra be quantified and added to the figures? A related question concerns the difference between standard deviation (SD) and standard error on the mean (SEM) for a measured variable. Is it correct to say that the BAY method produces SEM and the WBG method produces SD so the spreads in Figures 5 and 6 show different but equally interesting statistical properties of the measurements?

3) The use of Monte Carlo simulations to study possible biases (systematics) due to instrumentation effects is excellent and thorough. The authors show how the radiance profiles are affected by the measurement rod and a possible void/air gap between the ice and the rod, and how these biases depend on snow properties and therefore location. The discussion of the effects on radiance profiles is thorough and persuasive. However, it would really help in understanding and quantifying these effects to also show how the measured absorption spectra are affected by these systematics. It is finally quantified in terms of absorption in Section 3.5 and Fig 16 but this could be done at every previous stage also. I would have liked to see accompanying plots that show how the measured absorption depends on rod, depth, void, snow properties, and even as a function of true absorption. In this way, one could quantify this as a systematic uncertainty on the measurements and add this as an error band.

Minor comments:

Section 2.2: Some questions about the data selection:

1) How exactly were the homogeneous zones selected? Not all profiles look perfectly linear in the selected (gray shaded) zones.

2) How was the absorption fitted in each zone? A linear fit over the entire zone, or averaged over shorter linear fits to adjacent subsets of readings?

3) Where both descending and ascending profiles used and treated the same? Did they yield consistent results or were there systematic differences?

4) What is the explanation for the often quite large non-homogeneous zones, not close to the surface? Sometimes the whole profile is discarded. What was wrong in those cases, other than that they did not look linear in a visual inspection? Stated slightly differently (and a bit more provocatively): if there was nothing known wrong with the snow in the discarded zones other than that the profile did not look linear, how do you know that the snow in the selected zones is suitable for this measurement?

Page6Line31+: With the BAY method, the authors chose as prior a normal (in log scale) distribution with the average between IA1984 and IA2008 as the mean and as standard deviation the difference between the two (plus an extra SD factor for longer wavelengths where the two estimates agree). Leaving aside the impact of this choice on the result, the physical motivation seems somewhat flawed. The IA1984 estimate is based on lab measurements that are now known to be skewed (to stronger absorption) by scattering effects. The later AMANDA measurements showed that pure ice absorption must be much weaker than IA1984 and could even (in the absence of dust) be as weak as in IA2008. To use IA1984 to define the prior is therefore questionable. Given this objection, it would be relevant to see how much the choice of prior affects the measurement. Since the BAY results (Fig 6), which use this prior, end up close to the average WBG results (Fig 5) the effect of the prior is probably not too strong. However, what would happen if instead the difference between the weakest AMANDA absorption and IA2008 were used instead?

Page8Line27+: Isn't the good agreement at longer wavelengths (Fig 4) completely (not only "partially") explained by the methodology, i.e. that absorption is assumed to be known at 600 nm?

Page8Line31: If the empirical absorption model describing the AMANDA data holds beyond the deep ice, the spectral absorption shape is a combination of a falling power law due to dust absorption at shorter wavelengths and an exponential rise due to molecular absorption at longer wavelengths. The power law shape is fixed but the strength depends on dust concentration. This means that the cleaner the ice, the lower the power law part and the shorter the wavelength at the absorption minimum near the crossover point. This seems to be the trend in the measurements. There is no convincing evidence that any of the measurements are describing pure ice, so the minimum is not known. The minima in the measured spectra depend on dust contamination.

Page8Line34+: The description of the shape of the measured spectra is exactly that of the two-component model describing the AMANDA data. The small scatter at long wavelengths is because there the absorption of ice is measured, whereas at shorter wavelengths the absorption will depend on dust contamination. The results (Fig 5) confirm this picture. The result in Figs 7 and 8 further strengthen this interpretation, showing that the measured absorption depends on distance (and direction) from man-made activity at stations and therefore are most probably affected by dust contamination. In other words, the cleaner the ice, the weaker the absorption. This seems to confirm that dust is still a significant determinant of absorption below 450 nm in these data.

Page8Line37+: It is stated that the WBG absorption spectra (Fig 5) have different measurement quality and are thus not equiprobable. This is undoubtedly true, but uncertainties due to measurement quality should be separated from differences in spectra due to different dust contamination levels (because data is from different locations). In experimental results this "measurement quality" should be reflected in measurement uncertainty (error bars or bands). All spectra are shown as lines, without indicated uncertainty. Perhaps if measurement uncertainties (statistical and systematic) were

added, the spectra would all be consistent with the uncertainties? Probably this would be true for a given location but not between locations.

Finally, some minor language points:

In two places: a fiber optics -> an optical fiber

P13L31: back carbon -> black carbon

P10L37: neither -> either, nor-> or

---

## Author Comment (AC1) · 21 Sep 2016

Reviewer #1

This study presents new estimates of visible ice absorption coefficients, using a large number of measurements of light extinction in Antarctic snow. Precise estimates of the UV and visible absorption coefficient have eluded the scientific community because ice absorbs so weakly at these wavelengths, presenting numerous analytical challenges. This study builds on that of Warren and Brandt (2008) by (1) applying a larger set of measurements, (2) applying Bayesian statistical techniques that incorporate measurements at a larger number of wavelengths, and (3) conducting a rigorous modeling assessment of possible biases introduced by the presence of the fiber optic sensor and housing rod in snow. The current study finds that ice absorbs more strongly in the short-wavelength visible than found by Warren and Brandt (2008), though for unknown reasons. Overall, this is an important and very thorough study, certainly worthy of publication. The assessment of potential sensor biases using 3-D Monte Carlo modeling seems particularly rigorous and informative, and I think this aspect of the study will help inform past and future measurements of radiance extinction with depth in snow. I also expect that the new estimates of ice absorption spectra, available as supplementary data to the paper, will be widely used in the scientific community. It would have been satisfying if the authors had presented a convincing reason (or set of reasons) for why their estimates differ from those of Warren and Brandt (2008), but such an assessment may not be possible, or is otherwise beyond the scope of this study. It seems possible or likely that a combination of factors contributed to these differences. Below are some minor issues for consideration. Overall, I think this is an excellent study.

Minor comments:
p1,6: "larger than IA2008 by one order of magnitude..." - Larger at 400 nm, or averaged over the spectra? Please clarify.

Yes at 400nm and around. This is now indicated in the text.

p3,1-2: The inference from the quotation from Warren et al (2006), suggesting that more measurements should be made in snow further from sources of contamination, is that IA2008 measurements could be biased towards being too absorptive, but in fact this paper shows the opposite. Although this quotation is presented merely for motivation, reasons for the different findings between these studies appear to remain unknown. In general it would be helpful to offer (elsewhere in the paper) any additional insight or speculation that you have on reasons for the differences between these two studies. Convenient places for such discussion include sections 3.1, 3.4, and section 4.

We understand this request and Reviewer #2 had a similar one. It is indeed frustrating to obtain such a large difference and conclude this work without providing a satisfactory explanation. We tried to find one, though. For this we carried out a very detailed analysis of our protocol, so we were able to estimate the bias and uncertainty of our experiment (e.g. the effect of the rod, all the experimental details, etc). We also performed radiative transfer calculations to evaluate the possible impact of light absorbing impurities. Even if these evaluations are somewhat imprecise, our conclusion is that these errors are insufficient to explain the difference with IA2008. Conversely, we haven't performed a similar detailed analysis on Warren et al. (2006) protocol because we don't have enough information regarding their procedures. We have not identified any obvious flaw in the papers but undocumented details of their protocols could still be critical. Regarding AMANDA, the difference with our estimate is much smaller. It seems difficult to explain it as long as the difference to IA2008 remains unexplained.

Only the authors of these studies could reasonably reassess their error budget but we don't expect

much progress on this side because it is a difficult task.

We believe instead that it is easier and more promising to improve the experiments or design new ones to obtain complementary independent assessments. In this perspective, we hope that our paper, by showing up the problem, will motivate such future work where extra care on the protocol (and its description) will be taken. In this paper, we prefer to refrain us from adding speculations on others' experiments. We want to stick on the factual results we got.

Equation 1: Technically, "I(z=0)" should be "I(z=0,$\lambda$)" for consistency with the left-hand side of the equation.

done

p5,20: Please clarify what is meant by "the first two factors on the right hand side". The square root term makes this statement a bit ambiguous.

This is solved by reversing the sentence and explicitly naming the factors.

p5,19 and equation (3): It appears that the $\sigma$ terms refer to extinction coefficients of ice+air. If so, please explicitly clarify this. Otherwise, the distinction between $\sigma$ a and $\gamma$ ice is unclear, as both are absorption coefficients with identical units.

We have added "of the snow" after equation 2 and "$\sigma_s$ and $\sigma_a$ are the scattering and absorption coefficients of the snow." after equation 3.

Equation (3): It would be helpful to briefly explain the conditions under which the middle expression of Equation 3 apply, as it appears to be an approximation.

We have added:
"These equations assume that snow grains are randomly oriented particles, weakly absorbing and the real part of refractive index is nearly independent on the wavelength (Libois et al. 2013)."

p5,26: Does parameter B have much spectral dependence?

No, because it mainly depends on the real part of the refractive index and on the shape of the grains. We have changed the sentence as follows: "where $\rho_{ice}$=917 kg m$^{-3}$ is the ice density and B=1.6 the absorption enhancement parameter which has very little dependence to the wavelength.".

Equation (6): Please define the symbol $\alpha_n$ (perhaps accomplished most conveniently in the description of Equation 4).

It is implicitly defined by "All the variables (excepted zn ) are random variables". Equations 2 and 4 inspired the form of the equation 6 but the latter is not tight to a particular model of extinction. The only constrain on the model is the proportionality to the square root of the ice absorption. So, $\alpha_n$ is not related to Equation 4 and is nothing more than a "lump" proportional coefficient, i.e. a random variable with wide prior in Bayesian framework.

p6,17: "$\sigma$ measures the observation errors which are assumed identical for all the measurements." - Are there any conceivable or plausible conditions where the observation errors would depend on one or more key variables, such as depth in snow, surface irradiance, SSA, snow density, etc? In other words, what is the validity of this assumption?

It is likely that the error depends on various factors, especially the quantity of light. This implies a dependence to the wavelength , the depth and the density and SSA between the surface and the measurement depth. However it is quite complex because Solexs automatically adjusts the integration time to minimize the noise, so the relationship between depth and the noise level is not simple (and hardly predictable). It seems difficult to address such a complexity. In addition, adding more unknown variables to represent the dependence to the wavelength and depth would come at the price of an over-parametrization of the inverse problem with possible consequences on the convergence. We didn't try a more advanced scheme because we found that the posterior sigma is very small, indicating that the observation error is very small on average. It's also clear on the profiles in the supplementary figures that the measurements do not seem more noisy at larger wavelength or greater depth.

To highlight that this assumption is a simplification we slightly changed the sentence:
"σ measures the observation errors which, for sake of simplicity, are assumed identical for all the measurements."

p6,34: "The most likely value is taken as the average of these two parameterizations" - Do you mean that the available supplementary data are taken as the simple average of the WBG and BAY techniques, as applied to measurements collected from this study? Please clarify.

This sentence was linked to the previous one but a newline was inserted by mistake. We have reformulated the paragraph:

"The method WBG is indeed equivalent to considering that the prior of absorption is certain at $\lambda_0$=600 nm and ignorance at any other wavelengths. Here, we consider that IA1984 and IA2008 are two extreme  estimates (Warren, 1984; Warren and Brandt, 2008). Hence, the most likely prior value is taken as the average of IA1984 and IA2008 and the uncertainty related to the difference between IA1984 and IA2008 through a linear function. This function has an offset to represent the uncertainty when the difference is null, that is for $\lambda > 600$ nm. Note that since $\gamma(\lambda)$ spans several orders of magnitude in the visible, we compute difference and average in logarithm scale and choose a log-normal distribution for the prior.

p7,28: Presumably, the distribution of these step lengths is such that the extinction transmittance obeys Beer's Law. It could be worth mentioning nonetheless.

The Poisson random process is used to simulate the extinction events (either scattering or absorption) which is one of the processes of the radiative transfer. The Beer's law is different, it is an approximate solution of the radiative transfer valid for an absorptive or a slightly scattering medium (single scattering is sufficient), it does not apply to snow for which multiple scattering is significant.

Since using a Poisson process is common to many Monte-Carlo Radiative transfer models, we have simply added a reference.

p7,31: Briefly, how is bias avoided? Is it simply because the cutoff threshold is sufficiently small?

It's a refinement to avoid bias due to the cutoff whatever the threshold value. In principle we don't need it because we set the threshold to a value lower than the dynamics range of Solexs. This already guarantees that the bias (without Roulette) is small compared to the values we interpret. Nevertheless, using the roulette adds extra de-biasing. It was already implemented in MCML and is not expensive in terms of computation, we just keep it as it is.

We have added some explanations in the text:
Rays with an intensity less than a specified threshold are to be discarded. However, since abruptly discarding all the rays would result in a small bias (smaller than the threshold), a process known as Russian Roulette \(section 3.9 Wang et al. 1995) is applied. A small proportion of the rays (typically p=10%) is randomly chosen, their intensity is multiplied by 1/p and they are re-injected in the normal process of propagation. All the other rays are discarded. The threshold was set to $10^{-5}$ (the initial intensity of rays is 1) which ensures bias much smaller than SOLEXS dynamic range.

p8,15: Mode 2 (inverse tracking) is a creative solution to this problem.

It's a common technique in ray tracing to trace from the camera to the sources especially to account for diffuse radiation (large source, small sensor).

Section 3.3: This is a very informative and interesting analysis. It could also be very useful if you can offer any insight into how previous measurements of light extinction in snow should be re-interpreted, in light of this analysis.

We have inserted a comment at the end of section 3.3.2 (was 3.4 in the discussion manuscript) "It is worth mentioning that the problem of the rod absorption is evaluated here with a specific objective in mind but concerns any study exploiting measurements with inserted fiber optics  (King and Simpson, 2001; France et King, 2012). The interpretation of such measurements remains however highly tied to the specific protocol used."

12,31-32: This last sentence is unclear to me.

We have reformulated:
"This offset can be understood by the fact that the rod absorption is significant only over a few centimeters above the fiber tip, what was referred to as the "lower" part of the rod in Section 3.3.1. Hence, the gradient of the log-radiance is affected by the rod absorption as long as this lower part  is in a transition between two different homogeneous layers."

Section 3.5 and Figure 16: This analysis is unclear to me. Please consider revising the description of this sensitivity study for clarity. In particular, the determination of ice absorption using different ice absorption data (caption of Fig 16) seems circular. The third sentence of this paragraph (section 3.5) appears to explain the technique, but I don't quite understand what is being done here.

We have completely reformulated the section and the figure caption:

"The uncertainty range caused by the rod-snow interactions is evaluated here by considering the two snowpacks investigated earlier: the homogeneous snowpack which leads to an over-estimation of the AFEC and the 25kmE_1 snowpit which results in the opposite. To perform this evaluation, we ran MCML for each snowpack and for each ice absorption spectrum (IA2008 and BAY) which yields simulated radiance profiles as in Figures 9, 14, 15. We then apply the WGB method on these simulated profiles exactly as if they had been measured with SOLEXS. This yields the absorption spectra plotted in Figure \ref{fig_uncertainty_rod}. They ideally should be equal to the ice absorption spectra used as input for the simulations but  differ because of the rod absorption and the properties of the snowpack.

Figure 16} shows as expected that the homogeneous snowpack results in an over-estimation of the ice absorption and the opposite is true for the 25kmE_1 snowpack. The range between these two snowpacks is larger (1 order of magnitude) with the lower absorption spectrum (IA2008) than with

the BAY estimate (a factor 2 in linear scale) in the range 350—550 nm. In both cases, this uncertainty is larger than the statistical uncertainty estimated from the posterior (Section 3.2) which indicates that the rod absorption dominates the error budget. Nevertheless, the uncertainty ranges around IA2008 and BAY do not overlap which means that the rod absorption effect is insufficient to explain the discrepancy between both ice absorption estimates, at least considering that Warren et al. 2006 rod and snow properties were relatively similar to ours.

Figure caption:
\caption{\label{fig_uncertainty_rod}Ice absorption estimated with the WGB method applied to simulated irradiance profiles with MCML in the presence of the rod for the homogeneous snowpack (triangles up) and the 25kmE\_1 snowpit (triangles down) by using IA2008 (green) and BAY (black) ice absorption spectra, respectively, as input.}

13,28-30: This sentence should be fixed for clarity.

We have reformulated:
Although these spectra were obtained in different environmental conditions, we consider they are comparable because the ice absorption in the visible is known to be insensitive to pressure and the sensitivity to temperature is only of the order of +1% K$^{-1}$ (Woschnagg et al. 2001).

15,17-18: Could the large underestimations of ice absorption caused by rod-snow interactions be sufficient to explain the discrepancies in absorption between this study and Warren and Brandt (2008)? Insight on this would be helpful.

This question is addressed in the section 3.5 (now 3.3.3) which was unclear and has been reformulated. The answer to the first question is no, we don't believe the rod absorption can explain the difference, except if the rod used by Warren et al. 2006 was much much more absorbent than the one we use in the MCML simulations (which have been done for our rod). Only in case of a much more absorbent rod and a snowpack similar to Snow25E_1, the rod absorption could explain a very large under-estimation.. Moreover, if it were the case, the profiles of irradiance displayed in Warren et al. 2006 would probably be more irregular because of the dependence to the snow properties. All in all, this is very unlikely.

Figure 6 caption: What is the site of these measurements? Is it the same as in Fig 5?

yes, exactly the same data are used. We have updated the caption of Fig 6 based on that of Fig 5.

Figure 7: It is a bit difficult to match the colors of the lines to the legend, because they are so similar, but perhaps this is not important.

We have tried different color scales but the overlap of different colors resulted in grayish colors. We decided to add Figure 8 which is more informative. The Figure 7 is nothing more than a transition between Figures 5-6 and 8.

Figure 13: Needs a legend or color description in the caption to distinguish the two lines.

It has been added.

---

## Author Comment (AC2) · 21 Sep 2016

Reviewer #2

This paper presents updated estimates of the optical ice absorption spectrum in the visible based on a large set of new measurements of light extinction near the surface in Antarctic snow. The very weak optical ice absorption in the UV and visible creates experimental challenges which different techniques have been used to overcome. Lab measurements have been of limited use due to the required long path lengths and very pure ice samples. Measurements in deep natural ice in Antarctica by AMANDA (e.g., Ackermann et al., 2006) found that absorption in the visible is still dominated by dust contamination even for this extremely clean ice so intrinsic absorption by pure ice is experimentally hard to disentangle from absorption by impurities. Therefore, it has not been possible to definitely determine how strong optical absorption by pure ice actually is and existing measurements can in some sense only set upper limits, unless it can be shown that impurities and instrumental effects are negligible. Warren and Brandt (2008), referenced as IA2008, developed a technique to measure absorption through radiance measurements in highly-scattering Antarctic snow. Their measurement was in the end limited to a single snow layer, but they found an even weaker absorption than measured in deep ice by AMANDA. The current paper uses a similar technique as Warren and Brandt but improves on important aspects: a much larger data set, collected at different locations, is used; instrumental effects related to inserting the optical fiber assembly into the snow is studied with detailed 3D simulations; and they also use more sophisticated Bayesian statistical techniques to fit absorption parameters using all data at once. This work is very valuable in that it adds more information to the question of optical ice absorption near the minimum and also sheds more light on possible systematic uncertainties involved with such snow measurements. The paper should definitely be published, but I have some substantial comments on the current version.

1) The experimental technique, data collection, analysis, and bias discussion is thorough and described overall in a clear way (with some exceptions discussed below). My main comment concerns the interpretation of the result. Is the claim that these measurements arrive at an estimate of the absorption coefficient for pure ice, i.e. the intrinsic absorption by ice without impurities? It would be hard to make this case, considering previous measurements. We know from AMANDA that ice absorption which is still dominated by dust contamination is weaker than these new results. Therefore, the even weaker absorption in IA2008 logically comes closer to the true absorption for pure ice. The weakest absorption measured in AMANDA dips below 5×10-3 m-1, but this is still in ice with considerable dust and the spectral shape is the power law expected from absorption by dust. I would therefore expect pure ice absorption in the visible to be even weaker. The new BAY (clean) measurements show much stronger absorption than the cleanest AMANDA depths. We know that pure ice is at most as absorbing as the AMANDA ice. So the BAY estimate seems too absorbing. In this context it would also make sense to soften the definitive statement (Page13Line22) that "it is impossible to obtain absorption coefficient as low as IA2008". The large difference between the very weak IA2008 absorption and these new measurements with a similar technique is not understood, but logically IA2008 should be closer to true ice absorption since the AMANDA measurements set an upper limit.

The first and main goal of our study was to reproduce the Warren et al. experiment because its result is widely used by the snow modeling community. To this end we carried out many more measurements, used a better setup (e.g. finer resolution of the profile, fast execution of the measurements, …), and visited sites further from the Concordia station. The results show a much stronger absorption in the snow compared to Warren et al. 2006 which can not be explained by instrumental and statistical errors. This is the main topic of the result section. The first conclusion of the paper, at the end of the result section, is that the measurements of 2006 are not reproduced by the new experiment and the quotation Page13Line22 tells precisely this.

The second question indeed concerns whether the measured absorption is that of pure ice or not. This is addressed in the discussion section only, because we mostly refer to external results (literature review) to try answering this question. Our conclusion is that impurities content (BC or dust) measured at the surface of the ice-sheet are insufficient to explain the difference between IA2008 and BAY. At depth in the ice sheet AMANDA indeed faced the problem of dust during glacial periods, but modern time background concentrations are much weaker (e.g. dust are ~0.15ng/g eqBC) and could anyway not explain why IA2008 and BAY are different because both are subject to the modern time background. Only a huge increase of the pollution from the station, equivalent to 5ng/g BC could be responsible of such big difference. If the concentration at 25km South were large enough to affect our measurements, ie. 5ng/g eqBC, the concentration at the 4km would be at least (25/4)=6 times larger considering a uniform deposition rate and no effect of the wind. In practice it would be much more because our 25km South site is upwind. Six times is equivalent to 30ng/g eqBC. Such concentration would result in the following absorption (highest green curve):

[Figure]

The fact that AMANDA values are lower than the BAY values is not explained, and we do not have a sufficient knowledge of the details of the AMANDA's protocol and potential biases to further discuss this issue.

Based on all these results, it is more reasonable to consider that the various measurements available in the literature are biased or that the ice absorption has a hidden (large) sensitivity to some factor (e.g. temperature, …). The last conclusion of the paper is therefore that 1) more measurements with different techniques are still need, the subject is not closed and not simple and 2) meanwhile that considering that IA2008 is based on only one measurement and can not be reproduced, it is more reasonable to use BAY or AMANDA estimates, and to account for this uncertainty in critical calculations.

Regarding point 1), we have added the following at the end of the abstract: "future estimations of the ice absorption coefficient should also thoroughly account for the impact of the measurement method."

2) I would have liked to see the plotted data and simulation results shown with estimated uncertainties (error bars and bands) whenever possible to aid the interpretation. Can the statistical

and systematic uncertainties of the absorption spectra be quantified and added to the figures?
A related question concerns the difference between standard deviation (SD) and standard error on
the mean (SEM) for a measured variable. Is it correct to say that the BAY method produces SEM
and the WBG method produces SD so the spreads in Figures 5 and 6 show different but equally
interesting statistical properties of the measurements?

Several figures already show these errors. Figures 6 and 7 show set of curves drawn for the
posterior distribution, which is equivalent to error bands. Figure 8 also shows the marginal posterior
distribution for a given wavelength (a different view of Figure 7 results), where the width of the bell
shape curves is exactly the statistical error. The statistical error on the simulations with MCML are
not shown because there are weak. By using 10000 rays, the error is typically 1/sqrt(10000)~ 1%.
We have added the latter value (1%) in the text as it was missing. Figure 4 did not show statistical
errors. We have added this information (see the new figure below) and added a sentence in the text:
"In contrast, within this range, the statistical error (95\% confidence interval in gray shade) is very
small  which indicates the profiles have little noise and the estimation method is robust.". Figure 5
with confidence interval is also shown here but has not been inserted in the paper, for sake of clarity
and another reason evoked below.

[Figure]

[Figure]

Systematic error are more difficult to evaluate, as usual. Figure 16 shows the effect of the instrumental errors and Figure 17 shows the effect of pollution (green curves). Systematic error with MCML can be seen in Figure 9 by comparison with TARTES, it is very small.

As the review mentions, Figure 5 shows many curves which gives a rough idea of the SD but as explained in the text, all these curves have not the same accuracy, they can not be considered as equally valuable. Providing a detailed error analysis of the WGB is not the route chosen in the paper. We preferred to use the Bayesian framework because it is easier to model the errors and the inference/error propagation is more powerful. For instance we can choose the distribution of each variable and possibly even correlation between variable (but this was not done here), whereas the least square used in WGB assumes normal distribution for the dependent variable only. More importantly the Bayesian method really combines all these prior uncertainties of all the zones to produce the posterior. Roughly speaking, it means that thin and noisy homogeneous zones will be given a weaker weight than thicker zones with a perfectly linear log-irradiance profile. Hence, it is not false to say that the BAY method gives a sort of average – and the uncertainty shown in Figure 6 sort of SEM as suggested by the reviewer, but it is not mathematically equivalent.

3) The use of Monte Carlo simulations to study possible biases (systematics) due to instrumentation effects is excellent and thorough. The authors show how the radiance profiles are affected by the measurement rod and a possible void/air gap between the ice and the rod, and how these biases depend on snow properties and therefore location. The discussion of the effects on radiance profiles is thorough and persuasive. However, it would really help in understanding and quantifying these effects to also show how the measured absorption spectra are affected by these systematics. It is finally quantified in terms of absorption in Section 3.5 and Fig 16 but this could be done at every previous stage also. I would have liked to see accompanying plots that show how the measured absorption depends on rod, depth, void, snow properties, and even as a function of true absorption. In this way, one could quantify this as a systematic uncertainty on the measurements and add this as an error band.

It is true that the simulations with MCML are focused on the precise problem addressed in the paper which was 1) to help in the decision to discard the near-surface layers and 2) to evaluate the error on the final results for our specific setup (with our rod characteristics, ...).

The difficulty to address the request by the reviewer is two-fold:
1) this is complex to provide easily interpretable results (i.e. not misleading) because the snow properties interact with the rod characteristics (or air gap) and affect the estimated absorption spectrum. Therefore, for given rod characteristics, different absorption spectra can be obtained depending on the precise profile of SSA and density. This is clearly shown in Fig 16 (and associated text) with the homogeneous snowpack and 25kmE_1 snowpits. The homogeneous snowpack yields stronger absorption with more absorbent rod, and the 25kmE_1 yields the opposite. If we attempt to show the estimated absorption spectra resulting from variations of the rod radius or albedo, it would be strictly valid for the homogeneous snowpack and any extrapolation to another snowpack would be speculation. Therefore, it seems difficult to present general and interesting results without taking the risk to convey a misleading message. This may be better addressed in a dedicated study or, even better to our opinion, we recommend that future studies using fiber optics in snow should systematically use MCML-like simulations to evaluate the impact of the specific instrument used in the specific experimental conditions.

2) this request represents a significant amount of additional simulations as the sensitivity studies have been run at 400nm only. To plot ice absorption spectrum as in Fig 16 it would be necessary to run the same simulations for many wavelengths. Note also that the paper is richly illustrated with 17 figures.

Minor comments:

Section 2.2: Some questions about the data selection:
1) How exactly were the homogeneous zones selected? Not all profiles look perfectly linear in the selected (gray shaded) zones.

Each of the three authors performed an independent selection and the selections were automatically merged. Only zones for which two operators agree are retained (this is done with a script, there is no subjectivity in this part of the process). Each operator was recommended to remove the surface (up to 8cm), to reject zones where the ascending and descending profiles are too different and to select sufficiently thick portions of linear profile. Too thin zones would not affect the BAY method anyway since such zones would be "given" a low weight. This information is now provided in the text.

We have also provided  the profiles in supplementary Figure so the readers can reassess the dataset (and a digital version will be distributed here: http://lgge.osug.fr/~picard/ice_absorption/). Nevertheless, despite the subjectivity of this selection, it is worth noting that the final statistical error (Figure 6) is very small compared to the instrumental error, and the instrumental error is itself small compared to the big difference obtained between IA2008 and BAY estimates. It means that the selection of the zones has no impact on the main conclusions of the paper.

2) How was the absorption fitted in each zone? A linear fit over the entire zone, or averaged over shorter linear fits to adjacent subsets of readings?

The homogeneous zones are already quite thin (e.g. compared to Warren et al. 2006) so only a fit on each entire zone has been done. We changed the caption of Figure 5 to indicate this:
"70 ice absorption spectra estimated by the WBG method (Warren et al. 2006) on each homogeneous zone from the selection on the 56 radiance profiles measured around Concordia."

3) Where both descending and ascending profiles used and treated the same? Did they yield consistent results or were there systematic differences?

The data are combined before the estimation. It means that all pairs (depth, intensity) are taken independently of the direction (ascending or descending) of the acquisition. The direction is only used during the selection by the author to check the quality of the profile. In the supplementary figures, it is clear that the ascending/descending differ mostly by the level of intensity and not by the gradient, so the impact on the estimation should be small. Moreover, in such a case, because the data are not aligned, the profile will be "given" a small weight by the BAY method as it looks like noisy data.

4) What is the explanation for the often quite large non-homogeneous zones, not close to the surface? Sometimes the whole profile is discarded. What was wrong in those cases, other than that they did not look linear in a visual inspection? Stated slightly differently (and a bit more provocatively): if there was nothing known wrong with the snow in the discarded zones other than that the profile did not look linear, how do you know that the snow in the selected zones is suitable for this measurement?

In most cases, non-homogenous zones correspond to actual variations of the physical properties of the snowpack, there is nothing wrong with the measurements. Figure 14 shows that we reproduce the observed variations with MCML when the profiles of SSA and density ares taken into account. The WGB and BAY methods rely on the homogeneity of the zones (even if the instrumental errors were absent). This is a constraint of the methods.

For this reason Dome C is not an ideal site. With an annual accumulation of only 8 cm, the snowpack is usually very heterogeneous (Libois et al. 2014). A site combining high accumulation, weak wind and remoteness (for low impurities) would be ideal.

Page6Line31+: With the BAY method, the authors chose as prior a normal (in log scale) distribution with the average between IA1984 and IA2008 as the mean and as standard deviation the difference between the two (plus an extra SD factor for longer wavelengths where the two estimates agree). Leaving aside the impact of this choice on the result, the physical motivation seems somewhat flawed. The IA1984 estimate is based on lab measurements that are now known to be skewed (to stronger absorption) by scattering effects. The later AMANDA measurements showed that pure ice absorption must be much weaker than IA1984 and could even (in the absence of dust) be as weak as in IA2008. To use IA1984 to define the prior is therefore questionable. Given this objection, it would be relevant to see how much the choice of prior affects the measurement. Since the BAY results (Fig 6), which use this prior, end up close to the average WBG results (Fig 5) the effect of the prior is probably not too strong. However, what would happen if instead the difference between the weakest AMANDA absorption and IA2008 were used instead?

The arguments of the reviewer are relevant for the prior choice when an informative prior is wanted or needed. This is the case when 1) there is a strong reason to choose one particular value/spectra (here we really don't know which one is good) and 2) the posterior would be very large otherwise (here we obtain a narrow posterior). Choosing an informative prior is necessary as when a strong "regularization term" is needed in variational methods to stabilize the optimization. Here, we consider we don't know the value of absorption in the range 350-600nm, so we want a very large distribution for the prior of the ice absorption (i.e. uninformative prior), so that it does not influence the posterior. For this reason we have chosen IA1984 and IA2008 because they are extreme, they encompass the probable true value (as specified in the text). The choice was not driven by physical considerations or quality/flaw criteria. We could have chosen a uniform random variable between $10^{-5}$ and $10^{1}$ in the range 350-600nm without any reference to former ice absorption spectrum estimates, the results would have been essentially the same. This is visible because the posterior is much narrower than the prior. It means that the posterior is constrained by the data, not by the prior.

Page8Line27+: Isn't the good agreement at longer wavelengths (Fig 4) completely (not only "partially") explained by the methodology, i.e. that absorption is assumed to be known at 600 nm?

We don't understand the comment. The methodology imposes the value at 600nm so it influences the results at other wavelengths.

Page8Line31: If the empirical absorption model describing the AMANDA data holds beyond the deep ice, the spectral absorption shape is a combination of a falling power law due to dust absorption at shorter wavelengths and an exponential rise due to molecular absorption at longer wavelengths. The power law shape is fixed but the strength depends on dust concentration. This means that the cleaner the ice, the lower the power law part and the shorter the wavelength at the absorption minimum near the crossover point. This seems to be the trend in the measurements. There is no convincing evidence that any of the measurements are describing pure ice, so the minimum is not known. The minima in the measured spectra depend on dust contamination.

Page8Line34+: The description of the shape of the measured spectra is exactly that of the two-component model describing the AMANDA data. The small scatter at long wavelengths is because there the absorption of ice is measured, whereas at shorter wavelengths the absorption will depend on dust contamination. The results (Fig 5) confirm this picture. The result in Figs 7 and 8 further strengthen this interpretation, showing that the measured absorption depends on distance (and direction) from man-made activity at stations and therefore are most probably affected by dust contamination. In other words, the cleaner the ice, the weaker the absorption. This seems to confirm that dust is still a significant determinant of absorption below 450 nm in these data.

These two comments are related. We agree with the reviewer that there is a link in all the studies between the absorption value and the minimum of absorption. This is stated in our introduction: "A related debate concerns the position of minimum absorption, which in general has shifted towards shorter wavelengths with successive updates.". The interpretation with the AMANDA empirical model sounds possible. The decrease in the UV would be due to a power law due to small scatterers (Rayleigh scattering).

However, there is no evidence that dust is the cause of this scattering in our case (i.e. at the surface of the ice sheet). As written in the response to the major comments, the concentration of dust or soot needed to re-conciliate IA2008 and BAY is about 5ng/g eqBC, which, considering the literature, is impossible with the background (natural) concentration. Alternatively it would require a huge increase of pollution between 1km in ca. 2004 (IA2008 measurements) versus 25km upwind ca. 2012-2013 (our measurements). For comparison, recent chemical measurements at 1km from the Summit station in Greenland yields concentration between 50 and 300 ng/g (Carmagnola et al. 2013 in The Cryosphere) which is equivalent to about 0.3ng/g and 1.7 ng/g eqBC respectively. How concentration as high as 5ng/g eqBC could be systematically found at 25km upwind of Concordia ?

Calling at the dust to explain the discrepancy between the different estimates is not a more reasonable hypothesis than undiscovered instrumental errors, artifacts of protocol, or unknown sensitivity of the ice absorption to temperature, pressure, or crystalline structure.

Page8Line37+: It is stated that the WBG absorption spectra (Fig 5) have different measurement quality and are thus not equiprobable. This is undoubtedly true, but uncertainties due to measurement quality should be separated from differences in spectra due to different dust contamination levels (because data is from different locations). In experimental results this "measurement quality" should be reflected in measurement uncertainty (error bars or bands). All spectra are shown as lines, without indicated uncertainty. Perhaps if measurement uncertainties (statistical and systematic) were added, the spectra would all be consistent with the uncertainties?

Probably this would be true for a given location but not between locations.

The statistical error has been added on Fig 4 in the text . The same for Fig 5 has be done in this response. We have no robust way to estimate the systematic errors.

Finally, some minor language points:

In two places: a fiber optics -> an optical fiber

We have changed everywhere.

P13L31: back carbon -> black carbon
P10L37: neither -> either, nor-> or

done

---

## Author Response (AR2)

Dear Editor,

We are grateful for your decision and the precise suggestions to address the problem raised by the reviewers. Overall these suggestions are indeed a good compromise between informing the reader about the difference between IA2008 and our estimate (which was the reviewers request) and not over-concluding based on weak facts (which was our main concern and the reason why we only slightly changed the text).

Based on this, we have restructured the discussion to make clearer that we have explored different plausible artifacts but we also have not been able to quantitatively explain the discrepancy. We have also mentioned the impurities in the conclusion and abstract as suggested.

We provide the corrected manuscript with the differences highlighted in these three sections (abstract, discussion and conclusion).

Other minor changes have been implemented as suggested.

With best regards,
Ghislain Picard, Quentin Libois, Laurent Arnaud

[revised manuscript text omitted]